# Observation of quantum many-body effects due to zero point fluctuations in superconducting circuits

Sébastien Léger[1], Javier Puertas-Martínez[1], Karthik Bharadwaj[1], Rémy Dassonneville[1], Jovian Delaforce[1], Farshad Foroughi[1], Vladimir Milchakov[1], Luca Planat[1], Olivier Buisson[1], Cécile Naud[1], Wiebke Hasch-Guichard[1], Serge Florens[1], Izak Snyman[2] & Nicolas Roch[1]*

Electromagnetic fields possess zero point fluctuations which lead to observable effects such as the Lamb shift and the Casimir effect. In the traditional quantum optics domain, these corrections remain perturbative due to the smallness of the fine structure constant. To provide a direct observation of non-perturbative effects driven by zero point fluctuations in an open quantum system we wire a highly non-linear Josephson junction to a high impedance transmission line, allowing large phase fluctuations across the junction. Consequently, the resonance of the former acquires a relative frequency shift that is orders of magnitude larger than for natural atoms. Detailed modeling confirms that this renormalization is non-linear and quantum. Remarkably, the junction transfers its non-linearity to about thirty environmental modes, a striking back-action effect that transcends the standard Caldeira-Leggett paradigm. This work opens many exciting prospects for longstanding quests such as the tailoring of many-body Hamiltonians in the strongly non-linear regime, the observation of Bloch oscillations, or the development of high-impedance qubits.

[1] Université Grenoble Alpes, CNRS, Grenoble INP, Institut Néel, 38000 Grenoble, France. [2] Mandelstam Institute for Theoretical Physics, School of Physics, University of the Witwatersrand, Johannesburg, South Africa. *email: nicolas.roch@neel.cnrs.fr

The realization of many-body effects in quantum matter, often associated with remarkable physical properties, hinges on strong interactions between constituents. Mechanisms to achieve strong interactions include the Coulomb interaction in narrow band electronic materials, and Feshbach resonances that can produce arbitrarily large scattering lengths in ultracold atomic gases. In contrast, while providing great design versatility, purely photonic platforms[1–3] are not easily amenable to realizing strong correlations, since they usually come with weak nonlinearity. To circumvent this, superconducting circuits, which operate in the microwave range and display high tunability, have been proposed[4] for the exploration of correlated states of light. Here, correlations originate from nonlinear elements, such as Josephson junctions, and the enhancement of nonlinearities is accompanied by large zero-point fluctuations (ZPF). This can be understood in the electronics language of impedance as follows. The dynamics of a Josephson junction is described by two conjugate variables: the number of transferred Cooper pairs $\hat{n}$ and the superconducting phase difference $\hat{\phi}$. Despite being an anharmonic oscillator, a Josephson junction with Josephson energy $E_J$ and charging energy $E_c$, can be associated to an impedance $Z_J = \hbar/(2e)^2\sqrt{2E_c/E_J}$, which sets the amplitude of the fluctuations of $\hat{n}$ and $\hat{\phi}$. When $\langle\hat{\phi}^2\rangle$ is sufficiently smaller than unity, $\langle\hat{\phi}^2\rangle \propto Z_J/R_Q$ and $\langle\hat{n}^2\rangle \propto R_Q/Z_J$, with $R_Q = h/(2e)^2 \simeq 6.5$ kΩ the superconducting quantum of resistance. Consequently, at low $Z_J$, phase fluctuations are weak and the anharmonic Josephson cosine potential $E_J(1 - \cos\hat{\phi})$ can be reduced to a quadratic potential plus a quartic perturbation, as is the case for the transmon qubit[5]. On the other hand, if $Z_J$ is large, the full cosine potential is explored due to strong phase fluctuations. Anharmonicity then becomes important, as observed with the Cooper-pair box[6] or the fluxonium qubit[7], and as a result, the oscillation frequency $\omega_J$ can strongly deviate from the harmonic value $\sqrt{2E_JE_C}$. Thus, exploring many-body physics in circuit quantum electrodynamics must rely on a careful tailoring of ZPF.

The approach[8] that we follow to explore many-body effects originating from a single nonlinear superconducting element is to couple it to many harmonic modes. In the presence of such an environment, the degree of anharmonicity of a nonlinear junction will also depend on the external impedance, and three regimes can be identified. When $Z_J$ does not match the environmental impedance $Z_{env}(\omega)$ at frequencies $\omega$ close to $\omega_J$, the junction is accurately described as an almost isolated system, so that the effect of the environment only amounts to small perturbative corrections, similar to the Lamb shift[9]. At the same time, an impedance-mismatched environment remains weakly perturbed by the nonlinear junction, and this absence of back-action allows it to be described as a set of harmonic oscillators, following the Caldeira–Leggett approach[10]. This simplified description is at the core of the current understanding of open quantum systems, and was already verified experimentally in the early studies of macroscopic quantum tunneling[11]. The important role of ZPF in the damping effect that such an environment has on a Josephson junction was already noticed experimentally[12] and explained theoretically[13,14] three decades ago. In these early works, the effect of ZPF was to renormalize junction properties such as the critical DC current by about 1%, which nonetheless had a large effect on macroscopic quantum tunneling rates. When $Z_{env} \sim Z_J \ll R_Q$, the junctions and its environment fully hybridize, since they are impedance matched, but the anharmonicity of the junction remains weak and can be treated pertubatively[15–18]. The case $Z_{env} \sim Z_J \sim R_Q$ is much more challenging, both experimentally and theoretically since the strongly anharmonic junction hybridizes with many modes of its environment.

In DC measurements, such effects result in the celebrated Schmid–Bulgadaev transition predicted more than 30 years ago[19,20], a localization phenomenon whose relevance for microwave AC measurements requires further experimental and theoretical investigations[21]. The environment provides a strong action on the junction, which itself induces a sizeable back-action on many modes of the environment, the combined circuit forming a complex many-body system reminiscent of quantum impurity problems encountered in condensed matter[22]. More specifically, the frequency shift induced by the environment on the junction can be comparable with $\omega_J$, a nonperturbative effect due to a modification of the vacuum[23]. At the same time, the nonlinearity of the junction is transferred into the environmental modes, affecting for instance their broadening, and producing a physical regime that was not addressed so far.

In this work, we report on the effects of ZPF in a device consisting of a fully characterized multimode environment and a highly nonlinear single Josephson junction, acting as a weak link between two linear transmission lines, with all subsystems reaching the high-impedance regime. As a result, the transmission of single photons through our device is strongly affected by the interplay of nonlinearities and ZPF. We observe a 30% renormalization of the junction frequency as compared with the value that would have been obtained without ZPF—analogous to a giant Lamb shift—and we provide clear evidence for modifications of the environmental vacuum, which inherits strong nonlinear effects. A detailed temperature analysis of our system proves the quantum origin of these fluctuations and eliminates an explanation in terms of classical hybridization effects. Finally, our experimental findings are in quantitative agreement with a microscopic theory based on the self-consistent harmonic approximation (SCHA), embedded within a fully fledged microscopic description of our circuit using microwave simulation tools.

## Results

**Background**. The many-body regime of a single nonlinear junction coupled to a high-impedance environment has remained largely unexplored experimentally, since obtaining $Z_{env} \sim R_Q$ at gigahertz frequencies is very challenging. One option is to use on-chip resistors[24]. However, this may lead to unwanted Joule heating[25]. Therefore, we rather pursue a solution that relies on superconducting (lossless) high-inductance materials such as Josephson junction arrays[26–28], noting that disordered superconductors[29] are also promising. In Josephson junction arrays, $Z_{env} = \sqrt{L/C}$ can reach $R_Q$ given the large inductance of these materials, while maintaining good quality factors in the device.

Early experiments have embedded ultrasmall Josephson junctions between highly resistive leads, demonstrating the incoherent tunneling of Cooper pairs[24] in the framework of the $P(E)$ theory[30]. In this case however, no supercurrent flows through the junction and no quantum coherent effects were observed. Later, the phase/charge duality in the regime $Z_J, Z_{env} > R_Q$ was explored using SQUID arrays as the environment[31–33]. Experimental results were explained by fluctuations due to the finite temperature of the electromagnetic environment and the effect of ZPF could not be investigated. Moreover, these two series of experiments relied on DC measurements. This has the disadvantage that nonequilibrium effects need to be taken into account when results are interpreted, while the system is not directly probed at the finite frequencies—around $\omega_J$—that are of greatest interest.

It has since become possible to obtain a frequency-resolved picture of the environment of quantum systems such as Josephson junctions, thanks to the advent of circuit QED[34]. Here, microwave techniques allow a more accurate examination of the effects of ZPF on Josephson junctions[35], and observations

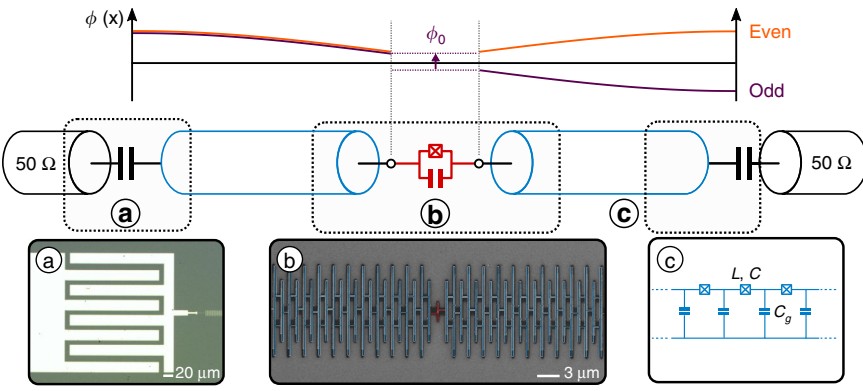

**Fig. 1** SQUID chains coupled to a small Josephson junction (weak link). The upper part represents the spatial phase distribution of the two first standing waves or resonant modes of the total system (Josephson junction + chains). And odd (even) mode—which couples (does not couple) to the junction—is represented in purple (orange). The lower part is a schematic of the system. The SQUID chains, depicted as blue transmission lines, are capacitively coupled to the input and output 50 Ω coaxial cables and galvanically coupled to the small Josephson junction (in red). **a** Optical picture of the input and output capacitive couplings. **b** SEM picture of a few of the SQUIDs (1500 in total for each chain) that are coupled to the small Josephson junction (in red). **c** Equivalence between the transmission line effective picture and the SQUID chain characterized by three microscopic parameters $L$ and $C$ the inductance and capacitance per SQUID respectively and $C_g$ the ground capacitance

**Table 1 Parameters of three samples. The bare Josephson energy $E_{J,bare}^{AB}$ is inferred using the Ambegaokar–Baratoff law. $E_J^*$ is the measured value of the renormalized Josephson energy. As a consistency check, the bare value $E_{J,bare}^{th}$ is also extracted from the fit of $E_J^*$ using the SCHA. $C_{sh}$ is the capacitance shunting the small Josephson junction (see Supplementary Note 9). $C$, $C_g$, and $L$ are obtained from the dispersion relation of the chain (see Supplementary Note 10)**

| Sample | A | B | C |
|---|---|---|---|
| Small junction | | | |
| Area [$\mu m^2$] | 315 × 195 | 370 × 190 | 440 × 185 |
| $C_J$ [fF] | 2.7 ± 0.3 | 3.2 ± 0.3 | 3.7 ± 0.4 |
| $C_{sh}$ [fF] | 3.0 ± 0.5 | 2.4 ± 0.4 | 5.1 ± 1.0 |
| $E_J^*$ [GHz] | 1.8 ± 0.1 | 3.1 ± 0.2 | 5.7 ± 0.3 |
| $E_{J,bare}^{AB}$ [GHz] | 3.7 ± 0.2 | 5.8 ± 0.3 | 6.8 ± 0.5 |
| $E_{J,bare}^{th}$ [GHz] | 3.7 | 5.5 | 8.2 |
| Nonlinearity $E_{J,bare}/E_c$ | 0.27 | 0.40 | 0.93 |
| Renormalization $E_J^*/E_{J,bare}$ | 0.49 | 0.56 | 0.70 |
| Chain | | | |
| $C$ [fF] | 144 | 144 | 144 |
| $C_g$ [fF] | 0.189 | 0.192 | 0.181 |
| $L$ [nH] | 0.66 | 0.60 | 0.61 |
| $E_J/E_c$ | 460 | 506 | 498 |

divergences in multimode models. Furthermore, a thorough modeling of such circuits is mandatory to discriminate the trivial effects of normal mode splitting (spectral shifts observed when two classical harmonic oscillators hybridize) from the dynamical ones associated to true vacuum fluctuations. With the exception of Gely et al.[47], this important issue has received surprisingly little attention in the circuit-QED context.

**Presentation of the experiment.** Our system builds on recent advances in the fabrication and control of large-scale Josephson arrays[18,48]. It consists of a small Josephson junction of characteristic impedance on the order of $R_Q$ ($E_J/E_c \lesssim 1$), which is embedded in the middle of two SQUID chains, each consisting of 1500 unit cells (Fig. 1), forming high-characteristic impedance transmission lines. We measure the characteristics of this environment precisely: its high-frequency cutoff—or plasma frequency—$\omega_{plasma} \simeq 17\,\text{GHz}$ and its wave impedance $Z_{chain} = \sqrt{L/C_g} \simeq 1.8\,\text{k}\Omega$ (see Table 1 and Supplementary Note 10). The SQUID parameters were carefully chosen to maintain a negligible phase slip rate ($E_J/E_c \lesssim 500$), ensuring that these chains can be described as a linear environment. They are capacitively coupled to the measurement setup to suppress DC noise which could affect the small junction (Fig. 1c). In order to vary the degree of nonlinearity and hence the strength of the ZPF, we measured three samples with different small junction sizes, connected to nominally identical chains (see Table 1).

The broadband microwave transmission of the full system shows a series of resonances (see Fig. 2b). A broadening of the modes in the array is expected since the SQUID chains are capacitively coupled to the 50 Ω measurement lines, hence forming very long microwave resonators. The transmission of the system is measured using very low microwave power, down to the single-photon regime. This prevents any power-induced broadening or frequency shift of these resonances (see Supplementary Note 9). A closer look at Fig. 2b reveals that resonances come in pairs. This is expected given the symmetry of the sample: our system can be decomposed into two subsystems (See Supplementary Note 2). One is made of even modes, which are decoupled from the small Josephson junction, while the other is composed of odd modes, with impedance $Z_{env} = 2Z_{chain}$, ultrastrongly coupled to the small Josephson junction[18,48]. A more surprising observation is that the odd modes are much more damped than the even ones. We interpret this as resulting

of perturbative spectral shifts (below 1%) attributed to ZPF were reported[9,36,37]. Several bottom-up experiments explored nonperturbative effects of light–matter interaction at ultrastrong coupling between a qubit and a single-mode resonator (for a review see Forn-Dìaz et al.[38] or Kockum et al.[39]). An effect similar to the Lamb shift—a reduction of the effective Josephson energy—was also reported recently for a DC-biased Josephson junction coupled to a single mode high-impedance resonator[40]. Moving towards many-body territory, a nonperturbative renormalization of the frequency of a flux qubit was demonstrated[41,42]. However, in this experiment, fluctuations were mainly thermal, and in addition, the environment cutoff frequency could not be clearly measured. The resulting unknown parameters prevented a quantitative modeling of the experiments. Indeed, as pointed by various authors[43–46], it is necessary to account for all the microscopic details of the circuit to get rid of unphysical

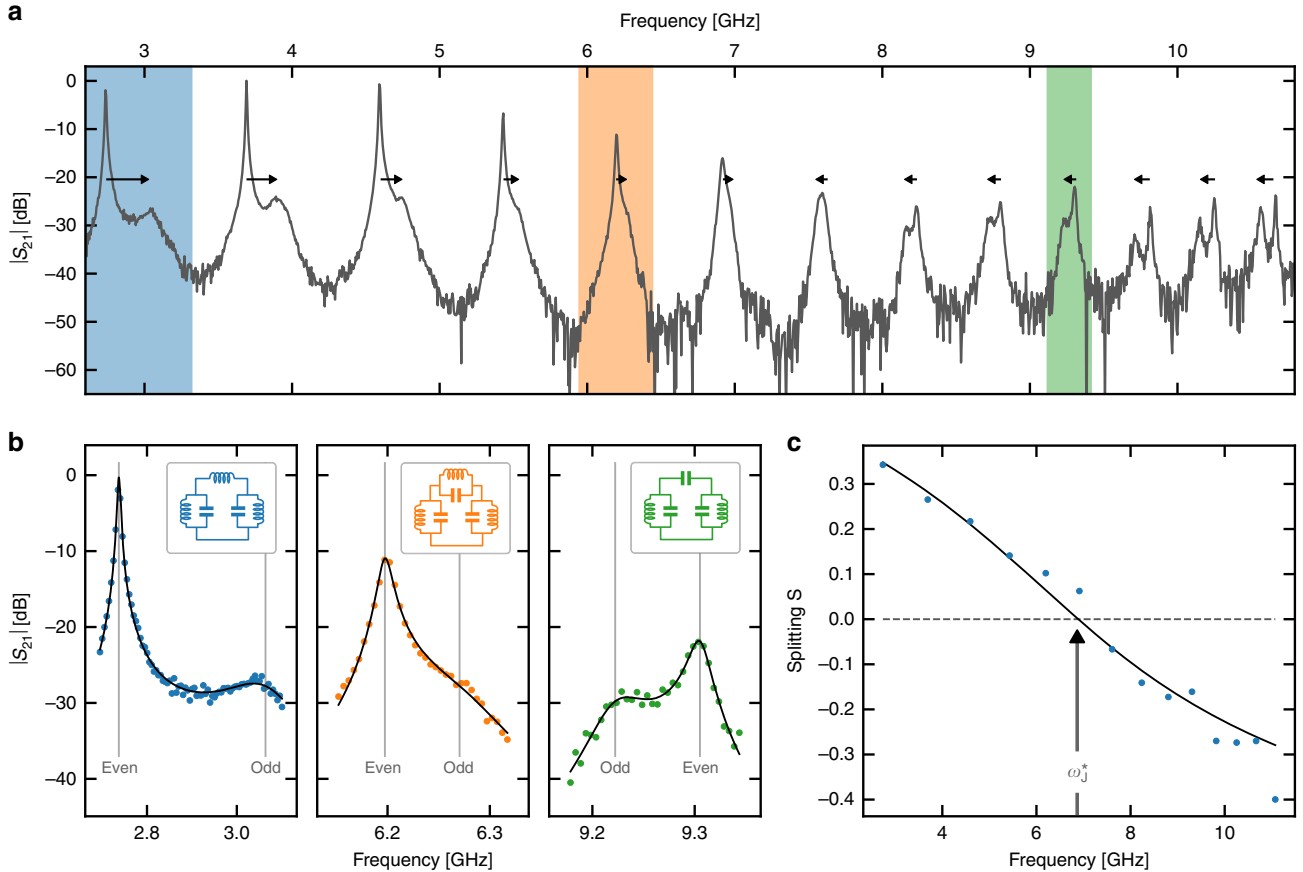

**Fig. 2** Inferring the renormalized resonant frequency $\omega_J^*$ of the small Josephson junction. **a** Amplitude of the microwave transmission $|S_{21}|$ versus frequency (sample A, 24 mK). The even–odd modes frequency splitting $S$ changes sign precisely at $\omega_J^*$. Arrows are guides to the eye of the splitting sign. **b** Fit of the double peaks for three cases: well below the resonant frequency of the small Josephson junction (blue) its inductive part dominates, close to $\omega_J^*$ (orange) the impedance of the junction is large so that the two modes are almost decoupled, and well above $\omega_J^*$ (green) the capacitive part of the junction dominates. **c** Experimental normalized frequency splittings $S$ obtained from the previous fits (dots) and theoretical prediction (full line). The resonance frequency $\omega_J^*$ of the small Josephson junction corresponds to the vanishing value of the normalized splitting $S$

from the nonlinearity that odd modes inherit from the small Josephson junction. This is experimental evidence of the strong back-action of the small Josephson junction on the many modes of the chain forming its linear environment.

### Line shapes

The line shape of a given even–odd pair of resonances can be obtained by associating with it two effective LC oscillator[49] connected via the small Josephson junction (see insets in Fig. 2b). In the regime of interest (See Supplementary Note 5 and 6) this junction can be treated as a ZPF-dependent inductance $L_J^*$ in parallel with a capacitance $C_\parallel$, with resonance frequency $\omega_J^* = 1/\sqrt{L_J^* C_\parallel}$. The odd and even modes mentioned earlier are characterized by respective frequencies $\omega_{\text{even}} = 1/\sqrt{LC}$ and $\omega_{\text{odd}} = 1/\sqrt{L_\Sigma C_\Sigma}$, with $1/L_\Sigma = 1/2L + 1/L_J^*$ and $C_\Sigma = C/2 + C_\parallel$. Then for modes at frequencies such that $\omega_{\text{odd}}, \omega_{\text{even}} \ll \omega_J^*$, the capacitance of the small junction can be neglected ($C_\parallel \sim 0$) leading to $\omega_{\text{odd}} > \omega_{\text{even}}$. In the opposite case ($\omega_{\text{odd}}, \omega_{\text{even}} \gg \omega_J^*$) the inductance can be neglected, giving $\omega_{\text{odd}} < \omega_{\text{even}}$. The most interesting regime is when the system is probed close to $\omega_J^*$. In that case, the impedance of the small junction diverges and consequently the two effective oscillators are uncoupled leading to $\omega_{\text{odd}} = \omega_{\text{even}}$. In the Supplementary Note 7, we confirm that a fully microscopic model of the whole circuit also predicts that the frequency splitting between even and

odd modes changes sign at the renormalized frequency of the junction $\omega_J^*$. The frequencies of each even–odd pair of modes is extracted by fitting the peaks Fig. 2 to line shapes of an input–output formalism based on the simple model just described (see Supplementary Fig. 3 and "Methods" section).

**Renormalized Josephson energy $E_J^*$.** The effective resonance frequency of the junction, $\omega_J^*$, depends on its environment due to the interplay of strong anharmonicity and many-body ZPF, and can be inferred by tracking the evolution of the normalized frequency splitting $S = (\omega_{\text{odd},k} - \omega_{\text{even},k})/(\omega_{\text{even},k+1} - \omega_{\text{even},k})$, between even (uncoupled) and odd (coupled) modes, where $k = 0 \ldots M$ refers to mode number. As shown in Puertas-Martìnez et al.[18], in a long chain, this quantity equals the phase shift difference between even and odd modes. It vanishes when the left and right halves of the device decouple, so that even and odd modes become degenerate. Figure 2c shows the experimentally obtained $S$ for one of our samples, from which we extract $\omega_J^*$. As we show in the Supplementary Notes 4 and 5, the ZPF-dependent effective inductance of the weak link is related to a renormalized Josephson energy $E_J^* = (\hbar/2e)^2/L_J^*$ as

$$\omega_J^* = \sqrt{2E_J^* E_c},\tag{1}$$

where $E_c = (2e)^2/(2(C_J + C_{\text{sh}}))$, with $C_J$ the intrinsic capacitance of the junction and $C_{\text{sh}}$ a shunting capacitance due to the

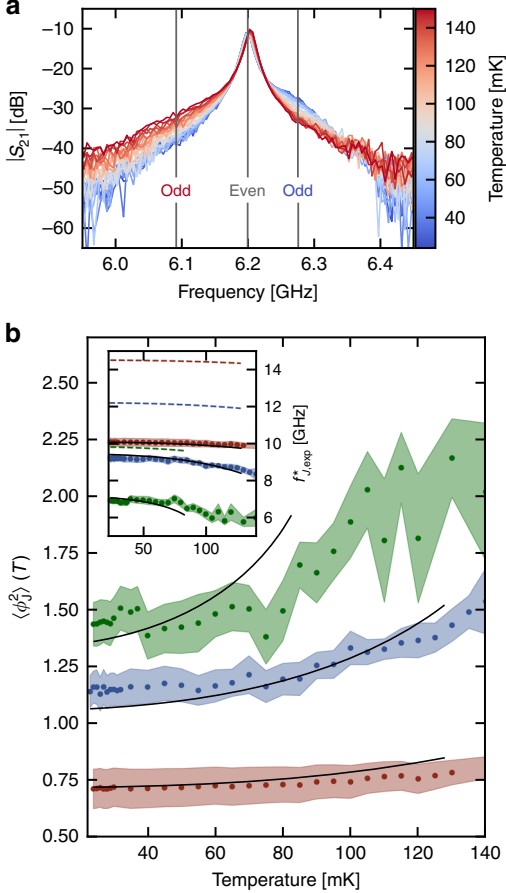

**Fig. 3** Temperature-induced renormalization. **a** Zoom on an even–odd pair of transmission peaks for sample A at temperature ranging from 23 to 150 mK. The even mode (gray) does not move while the odd mode (blue is at 25 mK, red at 130 mK) shifts down in frequency when warming up, showing a downward renormalization of the junction frequency $\omega_J^*$. **b** ZPF of the small junction $\langle \phi_J^2 \rangle$ as a function of the temperature for three samples (A, B, and C ranging from dark to light blue), extracted from Eq. (2). ZPF are stronger in sample A, which is associated to a smaller ratio $E_{J,bare}/E_c$ (large nonlinearity). The measured quantum to classical crossover is in good agreement with theory (full lines). The inset displays the corresponding renormalized junction frequency $f_J^* = \omega_J^*/2\pi$ of the three samples. The full lines are the SCHA predictions while the dashed lines represent what would be the temperature evolution of these frequencies if ZPF were omitted from $\langle \phi_J^2 \rangle$, using the same values of $E_{J,bare}$

surrounding circuitry. Note that we define $E_J^*$ in terms of $L_J^*$, and not in terms of the DC critical current as is done in for instance[23]. We use Eq. (1) to infer $E_J^*$ experimentally. $C_J$ is given by the junction size measured from an SEM picture. The way $C_{sh}$ is extracted is explained in the Supplementary Note 9. Values for sample A, B, and C are reported in Table 1. To see the effect of vacuum fluctuations, we compare $E_J^*$ with the bare Josephson energy of the weak link, which was obtained as follows. We fabricated many nominally identical Josephson junctions on the same chip and measured their room temperature resistances. The expected bare Josephson energy of the small Josephson junction $E_{J,bare}^{AB}$ (see Table 1) was then inferred using the Ambegaokar–Baratoff law. We observe a systematic shift between this bare energy and the renormalized one we inferred from $|S_{21}|$ measurements, a shift that is more pronounced for sample A that shows a high nonlinearity. This points towards a large renormalization induced by the strong zero-point phase fluctuations of the

hybridized junction-chain modes, as expected since the small junction is impedance matched to the chains.

We now show that this renormalization is quantitatively captured by a microscopic model based on the SCHA. Its success in accounting for nonlinearities introduced by Josephson junctions is well-established[13,14,50,51]. More recently it was employed in detailed microscopic models in the field of circuit QED[18,52]. The idea behind the SCHA is that the strong phase fluctuations allowed by the environment average the nonlinear potential of the small Josephson junction, lowering its effective Josephson energy from the bare value $E_J$ to the renormalized one $E_J^*$. This is valid, provided the phase $\phi_J$, though strongly fluctuating, is still sufficiently localized. In this regard we note the following. Though large, the effective environmental impedance $2Z_{chain} \simeq 3.8$ kΩ seen by the weak link, is still less than $R_Q$. Under this condition, the environment is known to produce spontaneous symmetry breaking of the $2\pi$ periodicity in the phase difference $\phi_J$ across the weak link[19,20,22]. It is therefore reasonable to approximate the system's full wave function with a Gaussian that is fairly well localized in the $\phi_J$ direction, which is the essence of the SCHA. At zero temperature, the interplay of many-body ZPF and nonlinearity can be described is approximated by replacing the cosine Josephson potential by an effective quadratic term $E_J^*\phi_J^2/2$, where the renormalized Josephson energy $E_J^*$ is given by the self-consistent equation:

$$E_J^* = E_{J,bare}\, e^{-\langle \phi_J^2(E_J^*)\rangle/2}. \tag{2}$$

Here, the total phase fluctuation across the junction $\langle \phi_J^2 \rangle$ is given by

$$\langle \phi_J^2 \rangle = \sum_{k\in odd} \phi_k^2, \tag{3}$$

where $\phi_k^2$ is the contribution to the small junction ZPF coming from odd mode $k$. Importantly, in the strong ZPF regime, the expectation value must be taken with respect to the modified vacuum of the hybridized modes, which means that the normal modes of the systems has to be updated during the numerical iteration of Eq. (2). This is in contrast to familiar examples of ZPF induced phenomena, such as the Lamb shift in hydrogen, where the perturbative nature of the effect allows one to calculate fluctuations with respect to the bare vacuum of the environment. We independently extracted the parameters of the whole circuit (junction + chains), and then used Eq. (2) to determine the theoretical bare Josephson energy required to find back the measured renormalized $E_J^*$ (see next section for more details). The agreement between experimentally and theoretically estimated $E_{J,bare}$ (see Table 1) provides strong evidence that our system displays large ZPF, which leads to a renormalization of up to 50% of the Josephson energy of the small junction (or equivalently 30% of its resonant frequency $\omega_J$). Moreover, as expected, this renormalization increases when the ratio $E_{J,bare}/E_c$ decreases, or equivalently when the nonlinearity of the small Josephson junction increases.

**Quantum versus thermal fluctuations**. As $\omega_J^*$ is renormalized by phase fluctuations across the weak link, one expects a crossover from quantum to thermally driven fluctuations as temperature increases. Extending the SCHA to nonzero temperatures (see Supplementary Note 4 and 5), we find that the fluctuations of mode $k$ contain a Bose factor contribution:

$$\phi_k^2(T) = \phi_k^2\left[1 + \frac{2}{\exp(\hbar\omega_k/k_B T) - 1}\right] \tag{4}$$

with $\omega_k$ the frequency of mode $k$, and $\phi_k^2$ its zero temperature ZPF. Therefore, at low temperature, fluctuations saturate to a

finite ZPF value (a hallmark of quantum uncertainty), while at high temperature they increase linearly with temperature ([4]). According to Eq. ([2]), $\omega_J^*$ should decrease when the system is heated up. Consequently, odd modes' frequencies are shifted to lower values when temperature increases, while the even modes stay put. This striking experimental signature of nonlinearity can clearly be seen in [3]a. This constitutes smoking gun evidence of the back-action of the Josephson junction on its environment: the shift of $\omega_J^*$ to smaller values at increasing temperatures indicates that fluctuations are thermally enhanced.

The recipe to extract $E_{J,bare}^{th}$ is the following: $E_J^*(T)$ is obtained from $S_{21}$ measurements at different temperatures. Since all the other parameters ($L$, $C$, $C_g$, $C_J$, and $C_{sh}$) are known, we can fit $E_J^*(T)$ using Eqs. ([2]) and ([4]), taking $E_{J,bare}^{th}$ as the (only) fitting parameter. Then, $E_{J,bare}^{th}$ being determined, we can compute the phase fluctuations across the small Josephson junction using Eq. ([1]) and ([2]):

$$\langle\phi_J^2\rangle(T) = 4\log\left(\frac{2E_{J,bare}^{th}E_c}{\omega_J^*(T)}\right). \tag{5}$$

We checked that at the lowest temperature of our cryostat, the phase fluctuations experienced by the small Josephson junction are fully in the quantum regime, by measuring $|S_{21}|$ from 25 to 130 mK. Results are shown in Fig. [3]b. We observe that the quantum to classical crossover appears at decreasing temperatures from sample C to A. This is because $\omega_J^*$ decreases from sample C to A. Therefore the junction is coupled to modes with lower and lower frequencies, which are thermally occupied at lower temperatures. The inset in Fig. [3]b shows the corresponding fit of $\omega_J^*$ for the three samples. The dashed lines represent $\omega_J^*$ obtained using the value of $E_{J,bare}$ extracted from the previous fit but including only thermal renormalization of $\omega_J^*$ i.e. disregarding ZPF. Consequently, $\langle\phi_J^2\rangle$ is given by:

$$\phi_k^2(T) = \phi_k^2\left[\frac{2}{\exp(\hbar\omega_k/k_B T) - 1}\right]. \tag{6}$$

The discrepancy between the dashed lines and the fit clearly shows that the fluctuations have mainly quantum origin. At increasing temperatures, thermal fluctuations add to the quantum ZPF, and cause a rise in $\langle\phi_J^2\rangle$, witnessed both in the experimental extraction and the predictions from SCHA, see Fig. [3]b. It is likely that the extracted $\langle\phi_J^2\rangle$ for sample A is systematically underestimated due to sizeable errors in the SCHA that rapidly set in after $\langle\phi_J^2\rangle \gtrsim 1$, leading to a mismatch with the theory at high temperatures.

**Many-body nature of the ZPF.** In order to confirm the many-body character of this renormalization, we can estimate how many modes are affecting the small junction simultaneously. The ZPF are quantitatively determined by how the full vacuum of the whole circuit is dressed by the coupling through the weak link. Within the SCHA, the number of modes contributing a finite amount of $\phi_k^2$ provides a measure of the number of interacting particles in the system. In Fig. [4] we compare the experimentally extracted $\langle\phi_J^2(T)\rangle$ with various calculated values. In each calculation, the full system was truncated to a finite number of modes in a window around $\omega_J^*$. If the window is too narrow, important contributions to the ZPF are neglected, and $\langle\phi_J^2(T)\rangle$ is underestimated. The comparison unambiguously shows that, in sample B, around 30 modes contribute to the total phase fluctuations. In circuit-QED language, the full width at half maximum (FWHM) of the environmental ZPF $\phi_k^2(\omega_k)$ – labeled $\Gamma_J$ – is about 7 GHz for our samples (see inset of Fig. [4]). Therefore, our device operates in a regime where $\Gamma_J/\omega_J^* \sim 1$ due to the impedance matching to the transmission line. Moreover, our device is

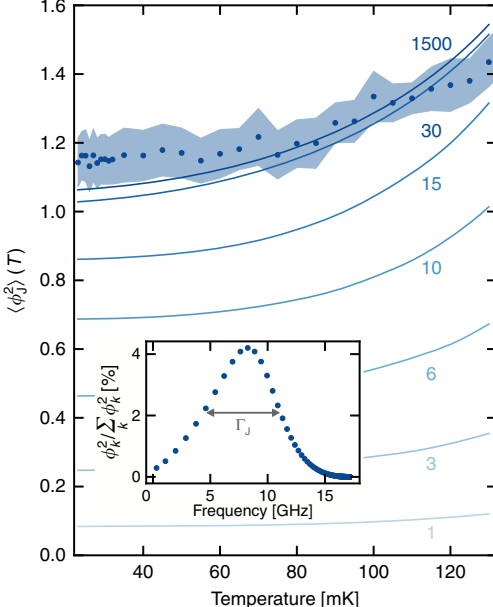

**Fig. 4** Many-body nature of the ZPF. Total phase fluctuations across the small Josephson junction in sample B, taking into account in our model (full lines) different numbers of modes of the environment, ranging from one (light blue) to the total number (dark blue). The inset shows the relative contribution of the different modes to the total fluctuations, with $\Gamma_J$ being the FWHM of this quantity

strongly nonlinear. Consequently, it is not possible to treat perturbatively the nonlinearity as is usually done in the field for the Transmon qubit or other weakly nonlinear circuits[5,15–17] (a detailed analysis is given in Supplementary Note 11).

This work provides a direct observation of several quantum many-body effects driven by ZPF in an open quantum system. This was achieved by developing a spectroscopic setup where the high-impedance environment of a single nonlinear Josephson junction was monitored mode by mode, and compared with a detailed microscopic model. A strong quantum renormalization (up to 50%) of the Josephson energy of the single junction was demonstrated, analogous to a nonperturbative Lamb shift. In addition, the back-action of the small Josephson junction causes nonlinear broadening and strong temperature dependence of the environmental modes, providing the most striking signature of the many-body effects that take place in our circuit. The measured temperature dependence of the phase fluctuation across the Josephson junction indicates that our device remains quantum coherent at cryogenic temperatures. As many as 30 modes are involved in the renormalization of the small junction. Our superconducting circuit thus behaves as a fully fledged quantum many body simulator, paving the way for the further observation of various many-body nonlinear effects in circuit QED[53–59].

## Methods
**Full model.** The Hamiltonian of the full system can be decomposed into odd and even parts—containing respectively the modes coupled and not coupled to the junction (see Supplementary Note 2). The odd Hamiltonian reads

$$\hat{H} = \hat{H}_0 + \left(1 - E_{J,bare}\cos\hat{\phi}_J\right), \tag{7}$$

$$\hat{H}_0 = \frac{(2e)^2}{2}\sum_{i,j=0}^{N}\hat{n}_i[\hat{C}]_{i,j}^{-1}\hat{n}_j + \frac{E_{J,S}}{4}\sum_{i=1}^{N-1}\left(\hat{\phi}_i - \hat{\phi}_{i+1}\right)^2 \tag{8}$$

with $\hat{n}_0 \equiv \hat{n}_J$ and $\hat{\phi}_J$ referring to the charge the phase drop across the small junction while $\hat{n}_i$ and $\hat{\phi}_i$, $i \in [1..N]$ refer to the charge and phase operators on chain site $i \in [1..N]$. Charge and phase operators obey the commutation rules

$[\hat{\phi}_k, \hat{n}_p] = i\delta_{k,p}$. The microscopic parameters are $E_{J,S}$, the Josephson energy of the SQUIDs, $E_{J,bare}$ the bare Josephson energy of the small junction, and the capacitance matrix:

$$\hat{C} = \frac{1}{2} \begin{bmatrix} C_I & -C & 0 & 0 & 0 & \dots & 0 \\ -C & 2C+C_g & -C & 0 & 0 & \dots & 0 \\ 0 & -C & 2C+C_g & -C & 0 & \dots & 0 \\ \vdots & \vdots & \ddots & \ddots & \ddots & \dots & 0 \\ 0 & 0 & 0 & -C & 2C+C_g & -C & 0 \\ 0 & 0 & 0 & 0 & -C & 2C+C_g & -C \\ 0 & 0 & 0 & 0 & 0 & -C & C_0 \end{bmatrix}$$

with:

$$C_I = 2(C_J + C_{sh}) + C + C_g, \tag{9}$$

$$C_0 = C_c + C_{c,I} + C. \tag{10}$$

**Self-consistent harmonic approximation (SCHA).** Because of the cosine term in Eq. (7), we are dealing with an interacting many-body problem that cannot be solved analytically. To study the best variational harmonic approximation we use the SCHA:

$$\hat{H} = \hat{H}_0 + \frac{E_J^*}{2}\hat{\phi}_J^2 + \left(1 - E_{J,bare}\cos\hat{\phi}_J\right) - \frac{E_J^*}{2}\hat{\phi}_J^2 \tag{11}$$

$$= \hat{H}_t + \left(1 - E_{J,bare}\cos\hat{\phi}_J\right) - \frac{E_J^*}{2}\hat{\phi}_J^2 \tag{12}$$

with $\hat{H}_t$ the trial harmonic Hamiltonian that will approximate $\hat{H}$, optimized with respect to the renormalized Josephson energy $E_J^*$. The variational principle gives

$$\frac{\partial}{\partial E_J^*}\langle\Psi_t|\hat{H}|\Psi_t\rangle = 0, \tag{13}$$

with $|\Psi_t\rangle$ the many-body ground state of $\hat{H}_t$. Because of the harmonic character of $\hat{H}_t$, we have

$$\langle\Psi_t|\cos\hat{\phi}_J|\Psi_t\rangle = e^{-\langle\Psi_t|\hat{\phi}_J^2|\Psi_t\rangle/2}. \tag{14}$$

Inserting (14) into (13) we end up with the self-consistent equation:

$$E_J^* = E_{J,bare}e^{-\langle\hat{\phi}_J^2(E_J^*)\rangle_t/2}. \tag{15}$$

The physical interpretation is the following: when ZPF are negligible, $\langle\hat{\phi}_J^2\rangle \simeq 0$ and $E_J^* = E_{J,bare}$, so in its ground state the junction behaves as an harmonic oscillator of frequency $\omega_{J,bare} = \sqrt{2E_{J,bare}E_c}$. For weak nonlinearity, fluctuations increase but remain such that $\langle\hat{\phi}_J^2\rangle \ll 1$, resulting in $E_J^* \simeq E_{J,bare}(1 - \langle\hat{\phi}_J^2\rangle/2)$, so that the junction behaves as a weakly anharmonic oscillator with fundamental frequency $\omega_{J,bare}(1 - \langle\hat{\phi}_J^2\rangle/4)$. For an isolated junction $\langle\hat{\phi}_J^2\rangle = \sqrt{E_c/2E_{J,bare}}$, and the frequency becomes $\omega_{J,bare} - E_c/4$, a well-known result for the Transmon qubit[5]. For larger fluctuations, the principle remains the same but no analytical formula can be derived, so that one should solve the self-consistent equation numerically. A more detailed derivation—including thermal fluctuations—is presented in Supplementary Notes 4 and 5.

**Frequency splitting S between odd and even modes.** The splitting $S$ is linked to the phase shift difference $\theta$ between even and odd modes in the thermodynamic limit[18]:

$$S = \frac{\theta}{\pi}. \tag{16}$$

The analytical formula of the phase shift difference—derived in the Supplementary 6 and 7 reads

$$\theta = 2\,\text{arccot}(X) + \arctan\left[\frac{1-\lambda}{1+\lambda}X\right] \tag{17}$$

with

$$X = \sqrt{\left(\frac{4C}{C_g}+1\right)\left(\left(\frac{\omega_p}{\omega}\right)^2 - 1\right)}, \tag{18}$$

$$\lambda = \frac{1 - \omega^2 CL}{1 + 2L/L_J^* - \omega^2 C_I L}, \tag{19}$$

$\omega_p = 1/\sqrt{L(C + C_g/4)}$ being the plasma frequency of the chain and $L_J^* = \hbar^2/(2e)^2 E_J^*$ the effective inductance of the small junction.

**Fitting formula for the peaks.** Input–output theory is used to fit the parameters associated to the double resonances observed in the transmission spectrum. These are mapped to two coupled harmonic modes $\alpha$ and $\beta$ with mutual coupling rate $g$, external coupling $\kappa_{ext}$ and internal loss $\kappa_{in}$, with Hamiltonian:

$$\hat{H} = \hbar\omega_r(\hat{a}_L^\dagger\hat{a}_L + \hat{a}_R^\dagger\hat{a}_R) + g(\hat{a}_L + \hat{a}_L^\dagger)(\hat{a}_R + \hat{a}_R^\dagger). \tag{20}$$

Here $\hat{a}_{in_L}$, $\hat{a}_{out_L}$ are the left input and output signals and $\hat{a}_{out_R}$ is the right output signal. Thus the input–output relations are :

$$\hat{a}_{in_L} + \hat{a}_{out_L} = \sqrt{\kappa_{ext}}\hat{a}_L, \tag{21}$$

$$\hat{a}_{out_R} = \sqrt{\kappa_{ext}}\hat{a}_R. \tag{22}$$

The equations of motion are:

$$-i(\omega - i\omega_r)\hat{a}_L + \frac{\kappa_{ext}}{2}\hat{a}_L = -ig\hat{a}_R - \sqrt{\kappa_{ext}}\hat{a}_{in_L}, \tag{23}$$

$$-i(\omega - i\omega_r)\hat{a}_R + \frac{\kappa_{ext}}{2}\hat{a}_R = -ig\hat{a}_L. \tag{24}$$

The complex transmission is defined as $S_{21} = \hat{a}_{out_R}/\hat{a}_{in_L}$, and can be calculated using Eqs. (21)–(24). We define the even $\omega_e = \omega_r + g$ and odd $\omega_o = \omega_r - g$ frequencies, and add phenomenologically losses in the odd modes $\kappa_o = \kappa_{in} + \kappa_{add}$ (we keep $\kappa_e = \kappa_{in}$), so that:

$$S_{21} = \frac{i\kappa_{ext}(\omega_o - \omega_e)}{(\kappa_{ext} + \kappa_o + -2i(\omega - \omega_o))(\kappa_{ext} + \kappa_e - 2i(\omega - \omega_e))}. \tag{25}$$

For some of the odd modes, we found a signature of inhomogeneous broadening, that we modeled by a convolution of their frequency with a gate function defined as $\Pi_{\delta\omega}(\omega) = 1/\delta\omega$ if $\omega \in [\omega - \delta\omega/2, \omega + \delta\omega/2]$. Understanding microscopically this additional broadening, possibly due to offset charges, is beyond the scope of the description using the SCHA, and will require additional theoretical developments.

## Data availability

The data that support the findings of this study as well the treatment scripts are available at the open access repository Zenodo [https://doi.org/10.5281/zenodo.3520349].

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

## Acknowledgements

The authors would like to thank F. Balestro, L. Del Rey, D. Dufeu, E. Eyraud, J. Jarreau, T. Meunier, and W. Wernsdorfer, for early support with the experimental setup. Very fruitful discussions with K. R. Amin, P. Forn-Diaz, J.-J. Garcia-Ripoll, D. B. Haviland, M. Houzet, P. Joyez, V. E. Manucharyan, F. Portier, and H. E. Tureci are acknowledged. The sample was fabricated in the Nanofab clean room. This research was supported by the ANR under contracts CLOUD (project number ANR-16-CE24-0005), GEARED (project number ANR-14-CE26-0018), by the National Research Foundation of South Africa (Grant No. 90657), and by the PICS contract FERMICATS. J.P.M. acknowledges support from the Laboratoire d'excellence LANEF in Grenoble (ANR-10-LABX-51-01). R.D. and S.L. acknowledge support from the CFM foundation and the "Investisements d'avenir" (ANR-15-IDEX-02) programs of the French National Research Agency. K.B. and J. D acknowledge the European Union's Horizon 2020 research and innovation programme under the Marie Skłodowska-Curie grant agreement No 754303.

## Author contributions

S.L., J.P.M., S.F. and N.R. designed the experiment. S.L. fabricated the device. S.L. performed the experiment and analysed the data with help from S.F., N.R. and I.S., while S.F. and I.S. provided the theoretical support. S.L., J.P.M., K.B., R.D., J.D., F.F., V.M., L.P., O.B., C.N., W.H.G., S.F., I.S. and N.R. participated in setting up the experimental platform, and took part in writing the paper.

## Competing interests

The authors declare no competing interests.

## Additional information

**Supplementary information** is avaliable for this paper at https://doi.org/10.1038/s41467-019-13199-x.

