## [Peer Review File · Nature Communications]

Reviewers' comments:

Reviewer #1 (Remarks to the Author):

In the manuscript, the authors demonstrate a strong renormalisation of the Josephson energy of a superconducting tunnel junction due to the collective effect of the quantum fluctuations of a large number of high-impedance engineered modes of its environment.

I find the manuscript strong, clearly written, and well supported by the observations and theoretical framework, and described sufficiently.

One comment: I would appreciate it if the authors would consider publishing the (optionally even raw) data, the data processing scripts, and the simulation code available on a public repository server (such as Zenodo): it would be tremendously valuable for the community and would significantly improve the usefulness of the work for the community by providing a unique and very powerful resource, inline with the new growing practices of open science. I know that not everybody does this, and by being among the first to do this makes the process a bit one-sided, and it may be perceived by the authors as sharing their valuable hard work while others keep it for themselves. But if the authors can set a good example, this can help swing the community towards being more open and sharing such highly valuable resources with each other, with the result that we can move together as a community in the development of the field.

Open data suggestions aside, it is my opinion that it very clearly satisfies the criteria of Nature Communications and am of the opinion that it should, in principle, be accepted for publication, after addressing the concerns below.

An aspect that I particularly appreciate in the manuscript is a correct description of the Lamb shift in which the precise role of zero-point fluctuations in the spectral shifts is identified, and I think the reporting of this beautiful experimental result has the potential to clarify some of the confusion in the literature regarding the origin of the Lamb shift in circuits.

In this regard, I have some concerns about the discussion of the literature in the manuscript in the context of the Lamb shift and quantum fluctuations: in particular, in the paragraph on page 2 that begins with "It has since become possible to obtain a...", it is not clear to me which of the previous experimental results the authors believe actually observed a non-perturbative Lamb shift, and which not? And furthermore, it is unclear if the Lamb shift that the authors are referring to in these previous results are due to quantum fluctuations or not? For example, the 1% that is referred to with respect to references 16 and 37: is that the "observed" spectral shift, or is that only the component of the spectral shift that is driven by quantum fluctuations? And similarly, in the open transmission line experiments: how much of their shift originates from quantum fluctuations? The authors are a bit vague on this.

I understand that the literature is confusing, but without a clear interpretation of the meaning of previous works, it is hard to determine the relevance of the work here. Unclear interpretation of previous work will propagate further the confusion in the literature, and it will undermine the recognition of this excellent result for what it is.

Also, along those lines, I find that the authors make a very important point at the end of this paragraph concerning the importance of separating the effects of quantum fluctuations from normal mode splitting (a topic also discussed in traditional quantum optics in the context of the contribution of radiation reaction in spontaneous emission).

However, the authors incorrectly infer, as far as I can see, that the importance of the separation of normal mode splitting effects (NMS) from those driven by quantum fluctuations (QF) is discussed in references 43, 44, and 46.

This statement surprised me, as, insofar as I know, the concept of separating out the contribution of zero-point fluctuations in the analysis is not discussed at all in references 43, 44, nor 46. After reading this statement, I read all of these papers again and was unable to find any mention or analysis of the separation of NMS and QF contributions in any of these references.

As far as I understand, the only reference that explicitly discusses and analyses for the separation of classical NMS and QF in the spectral shift (at least in recent circuit QED literature) is reference 45, and it is my opinion that the citations in the manuscript should be corrected to reflect this.

It is possible that I am incorrect in this, and that 43, 44, and 46 do indeed discuss and analyse the separation of classical NMS effects from those driven by QFs. If that is the case, I would ask the authors to point this out explicitly in their reply and specify the exact place in those manuscripts where this separation is made.

(As a side note, my interpretation is that 43, 44, and 46 perform an analysis of the convergence of multi-mode quantum Rabi models, a topic also discussed and analyzed in PRB 95 245115. Should the authors feel that citation of works about the convergence of multimode models are relevant for their manuscript, they could consider if a citation of PRB 95 245115 may also be relevant.)

A final comment I have is that although I believe the analysis that the quantum fluctuations result in a non-perturbative 50% renormalisation of the junction Josephson energy, I have difficulty identifying the experimental "smoking gun" signature of the effects of quantum fluctuations in their observations.

In particular, how would their experimental traces look say in figure 2 if quantum fluctuations were absent? Would the measurements be qualitatively different? Or would there be a quantitative shift? If there is a quantitative difference, how much difference would there be?

I understand, of course, that it is impossible to experimentally "turn off" quantum fluctuations. However, the authors have developed a detailed quantum model of their system: I guess it would be possible to just plot a theoretical prediction of what the model would give with the same parameters but with QFs set to zero by hand? Or by simply removing the anharmonicity of the small junction? This would give a clear immediate indication, for example, for the predicted relevance of QFs in their experimental observations.

In any case, it would make a much stronger case if the authors could make a concrete visual prediction of how their data would look like in the absence of quantum fluctuations.

Finally, I find the concept of non-anonymous peer review an interesting idea, and am glad that Nature Communications is exploring this. I have done my best to make my review above as objective and open as possible, and have thus chosen to reveal my name in the review and agree to publish my report and name online, should the authors choose for this. I hope the authors and community appreciate this gesture, and hope that this will result in a more open, transparent, and objective reviewing process for all journals in the future.

Reviewer Name: Gary Steele

Reviewer #2 (Remarks to the Author):

Various quantum properties of superconducting structures containing ultrasmall Josephson junctions (JJ) are being intensively investigated for decades, both theoretically and experimentally. Presently such systems attract a lot of attention of numerous researchers worldwide because of (i) their rich and highly non-trivial quantum behavior that can be directly tested in modern

experiments, in particular owing to a dramatic progress in nanofabrication techniques and (ii) a growing number of their applications, for instance in such fields as metrology and quantum information.

The manuscript under consideration reports on an experimental investigation of a fundamental issue of zero-point (or vacuum) fluctuations (ZPF) in single JJ embedded in an external quantum environment formed by two JJ chains with controlled parameters acting as effective transmission lines. This setup allows the authors to directly observe a significant (at the level of 30-50%) ZPF-induced decrease of the Josephson coupling energy of a single JJ as well as many-body and temperature effects.

The subject of the paper is definitely of interest to a broad readership, the experiment is accurately performed and clearly presented, the results of the observations are claimed to be in a good agreement with theory. At the same time, the present version of the manuscript suffers from several shortcomings among which I mention the following:

1. The authors seem to be unaware that the ZPF-induced renormalization of the Josephson critical current (manifesting itself, e.g., in a quantum shift of the flux in SQUIDs analogous to the Lamb shift) was predicted more than 30 years ago, see the papers by Zaikin and Panyukov, *Pis'ma Zh. Eksp. Teor. Fiz.* 43, 518 (1986) [*JETP Lett.* 43, 670 (1986)] and, in particular, (b) *Physica B* 152, 162 (1988). This prediction allowed, for instance, to fully eliminate the alleged discrepancy (by 5 orders of magnitude!) between theory and experiment on MQT in overdamped SQUIDs reported in (c) D.B. Schwartz et al. *Phys. Rev. Lett.* 55, 1547 (1985). With this in mind the work (c) can be considered as a first experimental manifestation of vacuum fluctuations in JJ. The authors should definitely include the abovementioned papers (a), (b) and (c) into the bibliography and give a brief overview of the issue in the introductory part of the manuscript.

2. The setup addressed in the paper is physically almost identical to that treated by Hekking and Glazman in Ref. [24]: In both cases the environment is represented by an effective transmission line and, hence, JJ interacts with a collection of Mooij-Schoen plasmons propagating along the chain/wire and acting as an effective Caldeira-Leggett bath of harmonic oscillators. In fact, essentially the same situation was also considered in the paper (b) except the Caldeira-Leggett bath was due to an external resistor (which should now be replaced by twice the impedance Z_{chain}). In both works (b) and [24] the Josephson coupling energy renormalized by ZPF was studied in details by means of different techniques. The authors should compare their experimental results with the corresponding theoretical predictions formulated in both (b) and [24] and report on the outcome in their manuscript. They should also comment on the relation between their own theory and those developed in the papers (b) and [24].

3. In the paragraph above Eq. (2) the authors introduce their key theoretical approach (SCHA) as if this approach would have been first invented in Refs. [22,49] which is, of course, by no means true. The idea to develop a variational technique approximating non-Gaussian interactions by an effective harmonic potential goes back to Feynman and had been employed by numerous researchers in various subfields of condensed matter theory long before Refs. [22,49]. In application to JJ the method of SCHA was used in the paper (b), just as one example (there are, of course, many others). In order to avoid possible misunderstanding the authors should modify their discussion of the SCHA technique accordingly as well as include proper references to this subject.

4. According to Eqs. (6)-(8) in the paper (b) quantum fluctuations influence not only the value of the critical current but also the form of the current-phase relation which now deviates from a simple sine form. Provided the critical current reduction is significant (e.g., in the range of 30-50% as in the present manuscript) these deviations are substantial as well.

Hence, the renormalized value of the critical current is now reached not at the phase value equal to $\pi/2$ but at some other value to be determined self-consistently. For the same reason, Eq. (1) for the Josephson plasma frequency does not apply anymore and needs to be corrected

accordingly.

5. In the introductory part of the manuscript the authors give a simple estimate for the averaged square of the phase fluctuations as being proportional to the ratio ZJ/RQ . Note that this estimate may only be appropriate under two conditions: (*) provided the Josephson phase is localized to one well, e.g., due to the so-called Schmid dissipative phase transition and, on top of that, (**) provided the strong inequality $ZJ \ll RQ$ is satisfied. The condition (**) is mentioned in the introduction in a somewhat misleading way that may create an illusion that the abovementioned estimate holds also if the inequality $ZJ \ll RQ$ is not satisfied. The condition (*) is not discussed by the authors at all. Moreover, in the text between Eqs. (14) and (15) the authors argue that their estimate for the averaged square of the phase fluctuations holds for an isolated JJ. This statement is fundamentally flawed since in equilibrium such JJ behaves as an insulator due to Coulomb blockade of Cooper pairs. Hence, the Josephson phase fluctuates strongly and the authors' estimate for phase fluctuations does not apply even qualitatively. Such fluctuations can strongly be reduced only in the presence of a low-impedance environment which crucial role appears to remain unclear to the authors.

To summarize, experimental results presented in the manuscript are valid, interesting and may constitute an important contribution to the field that could merit publication in Nature Communications. However, the present version of the manuscript contains a number of serious flaws and it cannot be accepted for publication for the reasons explained above. I suggest the authors to eliminate all the shortcomings mentioned in this report and after that to resubmit a revised version of their manuscript for further consideration.

Reviewer #3 (Remarks to the Author):

This work presents the quantum many-body effects of a single non-linear junction coupled to a high impedance environment- a demonstration of zero point fluctuations in an open quantum system. This work shows a strong quantum renormalisation of the Josephson energy of the single junction. The back action of the junction causes nonlinear broadening and strong temperature dependence of the environmental modes. The measured temperature dependence of the phase fluctuation across the junction shows 30 modes are involved in the renormalisation.

The many body regime of a single nonlinear junction coupled to a high impedance environment has first been investigated here. This work engineers the quantum environment, such that we can clearly see the effect of zero point fluctuations. The work is novel, interesting and could be published in Nature Communications, however, the authors need to address the following points.

In the middle of abstract (or first paragraph), "the resonance of the small junction acquires a frequency shift that is several orders of magnitude larger than for natural atoms." The references in natural atoms is missing. Details comparison with previous work is needed. If the author compares to Lamb shift, the Lamb shift in superconducting artificial atom has achieved 10 to 100MHz, this frequency shift of junction seems on the same order of magnitude. Please comment.

In page 2, first paragraph, fourth line, "observation of perturbative Lamb shifts...", I suggest to add the following collective Lamb shift work.

P. Y. Wen et al. "Large collective Lamb shift of two distant superconducting artificial atoms"
arXiv:1904.12473

In page 3, second paragraph, line 6, "The transmission of the system is measured using very low microwave power" The low power means comparing to the single photon power or what? What is

the single photon power defined?

In the caption of Fig. 1c, "1500 in total for each chain", how is the non-uniform of the junctions?
How does it affect the experiment?

In Fig.2 C, the unit of y axis S is missing, or it is normalised splitting instead of splitting?

#####

Reply to the first Referee

Reviewer #1 (Remarks to the Author):

In the manuscript, the authors demonstrate a strong renormalisation of the Josephson energy of a superconducting tunnel junction due to the collective effect of the quantum fluctuations of a large number of high-impedance engineered modes of its environment.

I find the manuscript strong, clearly written, and well supported by the observations and theoretical framework, and described sufficiently.

We thank referee number one for his positive assessment of our work.

One comment: I would appreciate it if the authors would consider publishing the (optionally even raw) data, the data processing scripts, and the simulation code available on a public repository server (such as Zenodo): it would be tremendously valuable for the community and would significantly improve the usefulness of the work for the community by providing a unique and very powerful resource, inline with the new growing practices of open science. I know that not everybody does this, and by being among the first to do this makes the process a bit one-sided, and it may be perceived by the authors as sharing their valuable hard work while others keep it for themselves. But if the authors can set a good example, this can help swing the community towards being more open and sharing such highly valuable resources with each other, with the result that we can move together as a community in the development of the field.

This suggestion makes complete sense. We believe as well that openness is a right direction for the future of science. Both our data and analysis scripts will be made available on Zenodo, once the paper is submitted to arxiv (according to Nature Communications Policy).

Open data suggestions aside, it is my opinion that it very clearly satisfies the criteria of Nature Communications and am of the opinion that it should, in principle, be accepted for publication, after addressing the concerns below.

Again, we thank the referee for stating that our work should be accepted for publication.

An aspect that I particularly appreciate in the manuscript is a correct description of the Lamb shift in which the precise role of zero-point fluctuations in the spectral shifts is identified, and I think the reporting of this beautiful experimental result has the potential to clarify some of the confusion in the literature regarding the origin of the Lamb shift in circuits.

In this regard, I have some concerns about the discussion of the literature in the manuscript in the context of the Lamb shift and quantum fluctuations: in particular, in the paragraph on page 2 that begins with “It has since become possible to obtain a...”, it is not clear to me which of the previous experimental results the authors believe actually observed a non-perturbative Lamb shift, and which not? And furthermore, it is unclear if the Lamb shift that the authors are referring to in these previous results are due to quantum fluctuations or not? For example, the 1% that is referred to with respect to references 16 and 37: is that the “observed” spectral shift, or is that only the component of the spectral shift that is driven by quantum fluctuations? And similarly, in the open transmission line experiments: how much of their shift originates from quantum fluctuations? The authors are a bit vague on this.

I understand that the literature is confusing, but without a clear interpretation of the meaning of previous works, it is hard to determine the relevance of the work here. Unclear interpretation of previous work will propagate further the confusion in the literature, and it will undermine the recognition of this excellent result for what it is.

Also, along those lines, I find that the authors make a very important point at the end of this paragraph concerning the importance of separating the effects of quantum fluctuations from normal mode splitting (a topic also discussed in traditional quantum optics in the context of the contribution of radiation reaction in spontaneous emission).

We agree that discriminating perturbative from non-perturbative Lamb-shifts, and spectral shifts from actual Lamb shifts (that are solely driven by vacuum fluctuations) is very important. As pointed out by the referee, this issue has been little discussed in the previous experimental literature on superconducting circuits. Moreover, one reads often some confusion between the effects of normal mode splitting (a purely classical effect) and of vacuum fluctuations on a quantum non-linear system, which in our opinion should be the genuine definition of the Lamb shift.

For example, in ref [16] the reported 1% spectral shift is mainly caused by normal mode splitting. As pointed in Physical Review A 98, 053808, vacuum fluctuations contribute to at most 10% of this spectral shift (the “true” Lamb shift is thus in the relative 0.001 range). In ref [37], these two contributions are referred to as static (normal mode splitting) and dynamic (vacuum fluctuations) Lamb shifts. According to the authors, the latter amounts to 1%, again a very small effect. Regarding the open transmission line experiments (we guess referee has in mind references [41] and [42]), a greater effect is observed, but it is caused mainly by thermal fluctuations, not zero point motion. So, in our opinion, the renormalization in Refs. [41-42] cannot be associated to a quantum Lamb shift, even though it also comes from the non-trivial interplay of fluctuations and non-linearity.

The main observation of our paper is a fully-quantum many-body, non-perturbative analogue of the Lamb shift. In the originally submitted manuscript, we had decided not to discuss in too much detail the difference between normal mode splitting and shifts driven by quantum fluctuations, as we wanted to highlight the difference between many-body and few-body effects. However, we now realize that this discussion should be carried out more thoroughly. We then updated this paragraph to reflect better the contribution of vacuum fluctuations to measured spectral shifts.

However, the authors incorrectly infer, as far as I can see, that the importance of the separation of normal mode splitting effects (NMS) from those driven by quantum fluctuations (QF) is discussed in references 43, 44, and 46.

This statement surprised me, as, insofar as I know, the concept of separating out the contribution of zero-point fluctuations in the analysis is not discussed at all in references 43, 44, nor 46. After reading this statement, I read all of these papers again and was unable to find any mention or analysis of the separation of NMS and QF contributions in any of these references.

As far as I understand, the only reference that explicitly discusses and analyses for the separation of classical NMS and QF in the spectral shift (at least in recent circuit QED literature) is reference 45, and it is my opinion that the citations in the manuscript should be corrected to reflect this.

It is possible that I am incorrect in this, and that 43, 44, and 46 do indeed discuss and analyse the separation of classical NMS effects from those driven by QFs. If that is the case, I would ask the authors to point this out explicitly in their reply and specify the exact place in those manuscripts where this separation is made.

(As a side note, my interpretation is that 43, 44, and 46 perform an analysis of the convergence of multi-mode quantum Rabi models, a topic also discussed and analyzed in PRB 95 245115. Should the authors feel that citation of works about the convergence of multimode models are relevant for their manuscript, they could consider if a citation of PRB 95 245115 may also be relevant.)

The referee is absolutely right. Only reference 45 analyses the relative contributions of NMS and QF. Our initial objective was to be as concise as possible to avoid overwhelming the reader with unnecessary technical discussions. This resulted in the merging of references discussing convergence problems in multi-mode light-matter interaction models (ref 43,44 and 46) with a reference highlighting the difference between NMS and QF (ref 45). The only common denominator of these papers is that they highlight the necessity to get a clear microscopic model of the circuit. This mistake of ours proves that the simpler is not always the better. We modified our paper to better reflect the discrepancies between the above-mentioned papers and cited PRB 95 245115. We thank the referee for his careful reading.

A final comment I have is that although I believe the analysis that the quantum fluctuations result in a non-perturbative 50% renormalisation of the junction Josephson energy, I have difficulty identifying the experimental “smoking gun” signature of the effects of quantum fluctuations in their observations.

In particular, how would their experimental traces look say in figure 2 if quantum fluctuations were absent? Would the measurements be qualitatively different? Or would there be a quantitative shift? If there is a quantitative difference, how much difference would there be?

This is a particularly important question. First, if either quantum fluctuations or the non-linearity were switched off, the broad odd peaks would sharpen to resemble the narrow even peaks. Having on the same device two similar sets of modes (odd and even) that respectively couple and decouple from the small junction provides a way to reveal qualitative effects triggered by the interplay of non-linearity and quantumness. Second, quantitative effects can be witnessed as well, although they always require a careful modeling and theoretical treatment of the circuit. For instance, the point where the splitting between even and odd modes changes sign (the junction frequency) is renormalized down to lower frequencies due to ZPF. While we cannot fully switch off the quantum fluctuations or the non-linearity, we do experimentally reveal their effect as follows: An independent determination of the bare E_J via a room temperature resistance measurement is shown to agree well with the bare E_J inferred from the measured renormalized E_J . Without a downward renormalization of junction frequency by ZPF, there would be a large discrepancy between these two estimates of the bare E_J . In addition, we show that when we increase phase fluctuations thermally, there is a further downward renormalization of the junction frequency. Our experimentally inferred temperature dependence of phase fluctuations does extrapolate to zero at $T=0$, but saturates at a finite value as T decreases.

Finally, we provide now some extra theoretical analysis in the supplementary information (see section J), which demonstrates that our device lies in a truly non-perturbative regime. In this respect, what sets apart our work w.r.t. most of the previous literature is the development of a microscopic and quantitative model, and the realization of a circuit displaying much bigger zero point fluctuations (resulting in a splitting by as much as 30%).

I understand, of course, that it is impossible to experimentally “turn off” quantum fluctuations. However, the authors have developed a detailed quantum model of their system: I guess it would be possible to just plot a theoretical prediction of what the model would give with the same parameters but with QFs set to zero by hand? Or by simply removing the anharmonicity of the small junction?

This would give a clear immediate indication, for example, for the predicted relevance of QFs in their experimental observations.

In any case, it would make a much stronger case if the authors could make a concrete visual prediction of how their data would look like in the absence of quantum fluctuations.

Indeed, removing the anharmonicity of the small junction can be done straightforwardly from our theoretical treatment.

As suggested by the referee, we modified figure 3 (see inset) to provide a clear visual description of QFs effect. We modified the associated caption and the text of our paper to emphasize this “smoking gun” signature. We thank again the referee for this suggestion.

Finally, I find the concept of non-anonymous peer review an interesting idea, and am glad that Nature Communications is exploring this. I have done my best to make my review above as objective and open as possible, and have thus chosen to reveal my name in the review and agree to publish my report and name online, should the authors choose for this. I hope the authors and community appreciate this gesture, and hope that this will result in a more open, transparent, and objective reviewing process for all journals in the future.

Reviewer Name: Gary Steele

We thank professor Steele for his open-mindedness.

#####

Reply to the second Referee

Reviewer #2 (Remarks to the Author):

Various quantum properties of superconducting structures containing ultrasmall Josephson junctions (JJ) are being intensively investigated for decades, both theoretically and experimentally. Presently such systems attract a lot of attention of numerous researchers worldwide because of (i) their rich and highly non-trivial quantum behavior that can be directly tested in modern experiments, in particular owing to a dramatic progress in nanofabrication techniques and (ii) a growing number of their applications, for instance in such fields as metrology and quantum information.

The manuscript under consideration reports on an experimental investigation of a fundamental issue of zero-point (or vacuum) fluctuations (ZPF) in single JJ embedded in an external quantum environment formed by two JJ chains with controlled parameters acting as effective transmission lines. This setup allows the authors to directly observe a significant (at the level of 30-50%) ZPF-induced decrease of the Josephson coupling energy of a single JJ as well as many-body and temperature effects.

The subject of the paper is definitely of interest to a broad readership, the experiment is accurately performed and clearly presented, the results of the observations are claimed to be in a good agreement with theory. At the same time, the present version of the manuscript suffers from several shortcomings among which I mention the following:

We thank the referee for his positive and critical assessment of our work.

1. The authors seem to be unaware that the ZPF-induced renormalization of the Josephson critical current (manifesting itself, e.g., in a quantum shift of the flux in SQUIDs analogous to the Lamb shift) was predicted more than 30 years ago, see the papers by Zaikin and Panyukov, *Pis'ma Zh. Eksp. Teor. Fiz.* 43, 518 (1986) [*JETP Lett.* 43, 670 (1986)] and, in particular, (b) *Physica B* 152, 162 (1988). This prediction allowed, for instance, to fully eliminate the alleged discrepancy (by 5 orders of magnitude!) between theory and experiment on MQT in overdamped SQUIDs reported in (c) D.B. Schwartz et al. *Phys. Rev. Lett.* 55, 1547 (1985). With this in mind the work (c) can be considered as a first experimental manifestation of vacuum fluctuations in JJ. The authors should definitely include the abovementioned papers (a), (b) and (c) into the bibliography and give a brief overview of the issue in the introductory part of the manuscript.

The referee is correct that ZPF-induced renormalizations of the Josephson energy have been reported before, and we have pointed to several experimental references already. In our view, our main contribution concerns the “many-body” character of the ZPF, as demonstrated firstly by the size of the effect in our device [a $\sim 50\%$ renormalization of E_J in our device as compared to $\sim 1\%$ in D.B. Schwartz et al. *Phys. Rev. Lett.* 55, 1547 (1985)] and secondly by the observation of the transfer of non-linearity (as discussed in Nigg et al, *PRL* 108, 240502) into many hybridized system-environment modes (revealed by the broadening of odd modes compared to even modes in our Figure 2). In the revised manuscript we added text to further clarify this. We also now include citations to the experimental (D.B. Schwartz et al. *Phys. Rev. Lett.* 55, 1547 (1985)) and theoretical (Zaikin and Panyukov, *JETP Lett.* 43, 670 (1986), *Physica B* 152, 162 (1988)) works suggested by the referee, which in addition to the already cited works (Schön and Zaikin, *Physics Reports* 198, 237 (1990), and Hekking and Glazman, *Phys Rev B* 55, 6551 (1997)) should clarify what precisely has been achieved prior to our contribution, both experimentally and theoretically. See the text at the top of the left column on page 2 of the revised manuscript.

2. The setup addressed in the paper is physically almost identical to that treated by Hekking and Glazman in Ref. [24]: In both cases the environment is represented by an effective transmission line and, hence, JJ interacts with a collection of Mooij-Schoen plasmons propagating along the chain/wire and acting as an effective Caldeira-Leggett bath of harmonic oscillators. In fact, essentially the same situation was also considered in the paper (b) except the Caldeira-Leggett bath was due to an external resistor (which should now be replaced by twice the impedance Z_{chain}). In both works (b) and [24] the Josephson coupling energy renormalized by ZPF was studied in details by means of different techniques. The authors should compare their experimental results with the corresponding theoretical predictions formulated in both (b) and [24] and report on the outcome in their manuscript. They should also comment on the relation between their own theory and those developed in the papers (b) and [24].

The referee raises here an interesting question. Our theoretical treatment is not fundamentally new per se, since the SCHA has been utilized by many authors (including the work by Zaikin and Panyukov quoted by the referee). What sets apart our theoretical analysis from all previous works that we are aware of in circuit-QED is that we take into account the complete microscopic details of our circuit. Indeed, although the environment can be described as Gaussian, the full capacitance network (see Fig. 2 in the supplementary material) has to be taken into account. For this reason, the spectral density of the bath is certainly not ohmic at the probed frequencies, and previous analytical formulas, derived under this assumption, cannot be used to obtain quantitative answers. But using the formalism of Zaikin and Panyukov with the proper Hamiltonian should give exactly the same results as ours (but this would require in any case a numerical calculation).

The work of Hekking and Glazman uses basically the same ohmic influence functional, but further takes the assumption of a charging energy of the weak link that is much larger than its Josephson energy, but this does not apply to our device. The result in Hekking and Glazman's paper that is analogous to our Eq. 2 is their Eq. 22. This latter formula gives an expression for the energy scale at

which the running relevant coupling becomes as large as the running ultraviolet cut-off. It therefore only determines the effective Josephson energy up to pre-factors of order one. Given all of this, it is not appropriate to quantitatively compare our measurements to the predictions of the two mentioned papers, although these pioneering works are certainly correct within the models they deal with. In fact, both these mentioned works predict a renormalization $\sim 10\% - 70\%$ for parameters similar to our device, which amounts indeed to the right order of magnitude.

3. In the paragraph above Eq. (2) the authors introduce their key theoretical approach (SCHA) as if this approach would have been first invented in Refs. [22,49] which is, of course, by no means true. The idea to develop a variational technique approximating non-Gaussian interactions by an effective harmonic potential goes back to Feynman and had been employed by numerous researchers in various subfields of condensed matter theory long before Refs. [22,49]. In application to JJ the method of SCHA was used in the paper (b), just as one example (there are, of course, many others). In order to avoid possible misunderstanding the authors should modify their discussion of the SCHA technique accordingly as well as include proper references to this subject.

We agree with the referee that the replacement of an anharmonic term with an effective harmonic term is a standard approximation that was explored in many previous works. In fact, our manuscript was directly transferred from Nature Physics (which requires up to 30 references maximum) to Nature Communications, and we should have updated our bibliography. We apologize if our bibliographic list was indeed missing some important references. Our choice of references was to present some works which develop detailed microscopic models in the field of circuit-QED. But we are very happy to include now a wider range of studies that fruitfully used similar calculations, namely :

- A. Kampf and G. Schön, Phys. Rev. B 36, 3651 (1987) [arrays of resistively shunted junctions]
- S. Chakravarty, G.-L. Ingold, S. Kivelson, and G. Zimanyi, Phys. Rev. B 37, 3283 (1988) [arrays of resistively shunted junctions]
- Zaikin and Panyukov, JETP Lett. 43, 670 (1986) and Physica B 152, 162 (1988) [a single resistively shunted junction]

See the paragraph above equation (2) in the revised manuscript.

4. According to Eqs. (6)-(8) in the paper (b) quantum fluctuations influence not only the value of the critical current but also the form of the current-phase relation which now deviates from a simple sine form. Provided the critical current reduction is significant (e.g., in the range of 30-50% as in the present manuscript) these deviations are substantial as well. Hence, the renormalized value of the critical current is now reached not at the phase value equal to $\pi/2$ but at some other value to be determined self-consistently. For the same reason, Eq. (1) for the Josephson plasma frequency does not apply anymore and needs to be corrected accordingly.

This is a very interesting question, and to our understanding, it seems that the referee's comment applies to the following situation. Let the renormalized Josephson energy of the weak link be defined via the critical current, i.e. $E_j^* = \hbar v Ic / 2e$, which means that it is determined from the part of the ground state energy vs. weak link phase difference where the slope is a maximum. The weak link resonance frequency on the other hand is determined by the energy-phase relation around its minimum. Only when the energy - phase relation is close to a cosine can the thus-defined effective Josephson energy and the weak link resonance frequency be related through our equation (1).

This estimate does not apply to our measurement, because we do not define or extract E_j^* from the critical current. We rather define E_j^* as $(\hbar v / 2e)^2 / L_j^*$, where L_j^* is the effective linear inductance of the weak link, within the SCHA. (See the text above our equation (1)). The content of our equation (1) is that the splitting between even and odd modes vanish at the resonance frequency of the weak link, which in the SCHA is given by $\sqrt{L_j^* (C_j + C_{sh})}$, a statement we prove in

Section F of the supplementary material. To make this clear we added the following sentence below equation (1). “Note that we define E_J^* in terms of L_J^* , and not in terms of the DC critical current as is done in for instance in Ref. [Hekking1997]”.

5. In the introductory part of the manuscript the authors give a simple estimate for the averaged square of the phase fluctuations as being proportional to the ratio ZJ/RQ . Note that this estimate may only be appropriate under two conditions: (*) provided the Josephson phase is localized to one well, e.g., due to the so-called Schmid dissipative phase transition and, on top of that, (**) provided the strong inequality $ZJ \ll RQ$ is satisfied. The condition (**) is mentioned in the introduction in a somewhat misleading way that may create an illusion that the abovementioned estimate holds also if the inequality $ZJ \ll RQ$ is not satisfied. The condition (*) is not discussed by the authors at all. Moreover, in the text between Eqs. (14) and (15) the authors argue that their estimate for the averaged square of the phase fluctuations holds for an isolated JJ. This statement is fundamentally flawed since in equilibrium such JJ behaves as an insulator due to Coulomb blockade of Cooper pairs.

Hence, the Josephson phase fluctuates strongly and the authors’ estimate for phase fluctuations does not apply even qualitatively. Such fluctuations can strongly be reduced only in the presence of a low-impedance environment which crucial role appears to remain unclear to the authors.

The picture given by the referee, while certainly correct, relies on a different experimental protocol than our measurement. Indeed, the referee described a DC current-biased (damped) Josephson junction, while we consider the equilibrium linear AC response of the system. In the former case, the phase is non-compact and subject to a washboard potential, so that a Schmid transition is crucial to localize the phase in one well. However, at equilibrium the superconducting phase remains compact, and behaves as a particle confined to a ring, subject to the cosine Josephson potential and to capacitive and inductive couplings to the rest of the system. When this particle is effectively localized to a portion of the ring (say around zero phase), and if excursions of the phase of order π are sufficiently unlikely, the SCHA can be applied. In this case, the ring can be replaced by the real line, and the cosine potential is replaced with a parabola, with the curvature of the parabola determined variationally. If the particle is confined to an interval much smaller than 2π , the SCHA becomes exact and the variationally determined curvature trivially becomes the bare E_J . For a less well-confined particle, the variationally determined curvature becomes significantly less than E_J , while the SCHA is still a good approximation. If however, the particle explores the full 2π interval, the results of the SCHA will become unphysical. We have checked that our device A lies in a regime where the SCHA is safe to apply (at zero temperature), since the phase fluctuations squared are about 1.3, much smaller than $\pi^2=10$. However, if the bare Josephson energy was much smaller than the one in sample A, the SCHA could not be trusted. To make this clear we have added the following qualification to our simple estimate in the introductory part « Provided that $\langle \phi^2 \rangle$ is sufficiently smaller than π^2 , $\langle \phi^2 \rangle \propto Z_J/R_Q$. » Below equation (2) we have also added a sentence « This is valid, provided $\langle \phi^2 \rangle$ is sufficiently less than π^2 . » Finally, we have also added the following sentence at the end of the Section « Quantum versus thermal fluctuations » : « We note that at low temperature the largest phase fluctuations are approximately $\langle \phi^2 \rangle \sim 1.3$, which is significantly less than π^2 , which justifies the use of the SCHA. A likely source of the discrepancy between theory and experiment at high temperatures, is the breakdown of the SCHA due to temperature enhanced phase fluctuations. »

The question about the isolated junction is an interesting one. In fact, by an isolated junction, we mean a junction shunted both by the junction capacitance and by the capacitance of the islands that it connects to. The effect of Coulomb blockade depends on the joint size of these capacitances. The estimate $\langle \phi^2 \rangle = \sqrt{2 E_c/E_{J, \text{bare}}}$ is well-known in the transmon qubit literature (an instance of which we cite) and holds quite well down to $E_J/E_C=1$, as can be verified from the exact solution for this problem. We do agree that the SCHA breaks down when the phase fluctuates

strongly, but our device never falls into this regime (except at elevated temperatures, due to extra thermal fluctuations).

To summarize, experimental results presented in the manuscript are valid, interesting and may constitute an important contribution to the field that could merit publication in Nature Communications. However, the present version of the manuscript contains a number of serious flaws and it cannot be accepted for publication for the reasons explained above. I suggest the authors to eliminate all the shortcomings mentioned in this report and after that to resubmit a revised version of their manuscript for further consideration.

We sincerely thank the referee for his critical comments, which were useful to improve the clarity of the manuscript.

#####

Reply to the third Referee

Reviewer #3 (Remarks to the Author):

This work present the quantum many-body effects of a single non-linear junction coupled to a high impedance environment- a demonstration of zero point fluctuations in an open quantum system. This work show a strong quantum renormalisation of the Josephson energy of the single junction. The back action of the junction causes nonlinear broadening and strong temperature dependence of the environmental modes. The measured temperature dependence of the phase fluctuation across the junction shows 30 modes are involved in the renormalisation.

The many body regime of a single nonlinear junction coupled to a high impedance environment has first investigated here. This work engineer the quantum environment , such that we can clear see the effect of zero point fluctuations. The work is novel, interesting and could be published in Nature Communication, however, the authors need to address the following points.

We thank the referee for his careful reading of our paper and appreciate that he recommends publication upon revisions.

In the middle of abstract (or first paragraph), “the resonance of the small junction acquires a frequency shift that is several orders of magnitude larger than for natural atoms.” The references in natural atoms is missing. Details comparison with previous work is needed. If the author compare to Lamb shift, the lamb shift in superconducting artificial atom has achieved 10 to 100MHz, this frequency shift of junction seems on the same order of magnitude. Please comments.

We added a reference to the seminal work of Lamb and Rutheford. Following the suggestions of both referee one and three, we expanded our discussion about previous literature regarding Lamb shift physics in superconducting artificial atoms. As pointed by referee number three, Lamb shifts as large as 100 MHz were reported before in cQED (ref 16 and 37). In our case this shift amounts to as much as 30% of the bare frequency of the artificial atom or equivalently to 3 GHz. This non-perturbative shift is a signature of the many-body state formed between the atom and its environment.

In page 2, first paragraph, four line, “observation of perturbative Lamb shifts...”, I suggest to add the following collective Lamb shift work. P. Y. Wen et al. “Large collective Lamb shift of two distant superconducting artificial atoms” arXiv:1904.12473

We thank the referee for pointing out this reference that we missed. It is now added to our manuscript.

In page 3, second paragraph, line 6, “The transmission of the system is measured using very low microwave power” The low power means comparing to the single photon power or what? What is the single photon power defined?

Low power indeed means in the single photon regime. As described in the supplementary material, modes shift when probed with increasing power. Experimentally we made sure that the power was low enough so that such power-induced shifts are negligible. We clarified this point in the body of our paper.

In the caption of Fig. 1c, “1500 in total for each chain”, how is the non-uniform of the junctions How does it affect the experiment?

According to our previous works, disorder is between 2% and 3% in the chains (see for example Javier Puertas, Probing light-matter interaction in the many-body regime of superconducting quantum circuits. Université Grenoble Alpes, 2018.). Such low disorder does not affect the experiment since the localization length remains much larger than our sample (see for example Basko, D. M. & Hekking, F. W. J. Disordered Josephson junction chains: Anderson localization of normal modes and impedance fluctuations. *Physical Review B* **88**, 094507 (2013).)

In Fig.2 C, the unit of y axis S is missing, or it is normalised splitting instead of splitting?

Fig 2.C indeed refers to normalised frequency splitting S. We thank the referee for pointing out this typo. We corrected figure 2 caption accordingly.

#####

List of changes

- 1) Data and analysis scripts related to this work will be available online as soon as our paper appears on the Arxiv.
- 2) According to referees one and three suggestions, we updated the paragraph discussing previous art regarding Lamb shift. We also elaborated on the difference between normal mode splitting and shifts purely caused by vacuum fluctuations. All these changes are highlighted in red.
- 3) As suggested by referee #1, we devised a new figure 3 to provide a visual representation of ZPF contributions to the Josephson energy renormalisation.
- 4) We modified the abstract by adding a reference to the Lamb shift observed for natural atoms, as suggest by referee #3.
- 5) We elaborated on our statement about low power measurements.
- 6) We corrected the typo in the caption of figure 2.c as suggested by referee #3.
- 7) References added:
 - M. Gely et al. PRB 95 245115 (suggested by referee #1)

- Zaikin and Panyukov, JETP Lett. 43, 670 (1986) and Physica B 152, 162 (1988) (suggested by referee #2)
- D.B. Schwartz et al. Phys. Rev. Lett. 55, 1547 (1985) (suggested by referee #2)
- P. Y. Wen et al. "Large collective Lamb shift of two distant superconducting artificial atoms" arXiv:1904.12473 (suggested by referee #3)

REVIEWERS' COMMENTS:

Reviewer #1 (Remarks to the Author):

I thank the authors for the careful considerations of my comments. I feel now that all of my concerns have been addressed and am happy to recommend the manuscript for publication in nature communications

Reviewer #2 (Remarks to the Author):

I have carefully read through both the revised version of the manuscript and the authors' rebuttals to my previous remarks. At a number of points the authors' responses are satisfactory. At those instances they improved both the presentation and the bibliography in a proper manner. Still, there remain a few issues where their text revisions and comments come incorrect and/or misleading.

1. In response to my critique in the Introduction section the authors have added the sentence: «Provided that $\langle \phi^2 \rangle$ is sufficiently smaller than π^2 , $\langle \phi^2 \rangle \propto Z_J/R_Q$. ». While the comment on the validity range of the above estimate for $\langle \phi^2 \rangle$ is indeed appropriate, the indicated condition $\langle \phi^2 \rangle \ll \pi^2$ is not quite correct. The estimate $\langle \phi^2 \rangle \propto Z_J/R_Q$ actually holds only provided $\langle \phi^2 \rangle$ remains (much) smaller than unity, whereas the number $\pi^2=10$ is totally irrelevant in this context. The authors should keep the above sentence in place but correct it accordingly, e.g., simply replacing « π^2 » by the word «unity».

2. Below Eq. (2) the authors have added the sentence: «This is valid, provided $\langle \phi^2 \rangle$ is sufficiently less than π^2 . » Furthermore, in the section «Quantum versus thermal fluctuations» below Eq. (6) they have added the paragraph: « We also note that at low temperature the largest phase fluctuations are approximately $\langle \phi^2 \rangle \sim 1.3$, which is significantly less than π^2 , which justifies the use of the SCHA. A likely source of the discrepancy between theory and experiment at high temperatures, is the breakdown of the SCHA due to temperature enhanced phase fluctuations... ». These text additions demonstrate that the authors do not fully understand the method of SCHA in general and its applicability range in particular. SCHA is a variational technique replacing an anharmonic potential by an effective harmonic one with some effective parameters to be determined self-consistently. Hence, within the framework of SCHA $\langle \phi^2 \rangle$ in Eq. (2) can in principle take ANY value ranging from zero to infinity. If, for instance, the latter limit is realized, i.e. $\langle \phi^2 \rangle$ tends to infinity, it would simply imply that (within SCHA!) the effective Josephson coupling energy E^*J gets renormalized down to zero by interactions. Hence, no ad hoc restrictions on $\langle \phi^2 \rangle$ exist within SCHA and the sentence right after Eq. (2) as well as the cited paragraph below Eq. (6) should be deleted as misleading. Perhaps I can also add that even following the authors' logics the discrepancies observed in Fig. 3b at higher temperatures could hardly be explained since for all three samples $\langle \phi^2 \rangle$ remains much smaller than 10 up to highest T.

3. Towards the end of the Introduction section the authors added the sentence «In DC measurements, such effects result in the celebrated Schmid-Bulgadaev transition predicted more than thirty years ago [27, 28], a localization phenomenon whose relevance for microwave AC measurements is presently debated [29]». The first part of this sentence is OK (apart from misspelling the name Schmid), while the second one (quoting a highly controversial preprint [29]) is problematic, since BY DEFINITION any quantum phase transition (QFT) including, of course, the localization QFT of the Schmid-Bulgadaev type, deals with the limit of both zero temperature and zero frequency. Hence, there is nothing to debate. I would strongly recommend the authors to leave out the second part of the above sentence in order to avoid any mentioning of microwave AC measurements in the context of the Schmid-Bulgadaev QFT.

4. The authors' reply to the comment #5 from my previous report is by no means satisfactory. In addition to an improper discussion of the SCHA validity range (already mentioned above) this reply contains a number of incorrect and misleading statements which I cannot avoid commenting on. For instance, the authors claim that «...at equilibrium the superconducting phase remains compact...». Or, they write that «The estimate $\langle \phi^2 \rangle = \sqrt{2 E_c/E_{J, \{\text{rm bare}\}}}$ is well-known in the transmon qubit literature... and holds quite well down to $E_J/E_C=1$, as can be verified from the exact solution for this problem». Unfortunately, these and some other related statements are in error. To begin with, the issue of being in or out of equilibrium has nothing to do with compactness or non-compactness of the superconducting phase. The latter is compact as long as the Josephson junction (JJ) charge remains discrete (e.g., quantized in units of $2e$) and gets decompactified if the charge of JJ becomes effectively continuous as a result of interactions with an external circuit. Furthermore, even if the phase is compact (i.e. defined on a ring), it does not imply that it is necessarily confined to the interval between zero and 2π , contrary to the authors' belief. The point is that the phase can take as many winds (rotations) around the ring -- both clockwise and counterclockwise -- as possible. As a result, e.g., in the true ground state of an isolated JJ and for ANY relation between EJ and EC the charge is always localized (thus being a c-number variable), whereas the phase suffers strong quantum fluctuations making $\langle \phi^2 \rangle$ diverge. It is just THIS behavior which follows from the exact solution of the problem no matter what «is well-known in the transmon qubit literature». In other words, an isolated JJ in or close to its ground state always demonstrates Coulomb blockade of Cooper pairs, thus being an insulator rather than a superconductor. In fact, this simple physics was well established and understood already three decades ago. For more details I can refer the authors, e.g., to the review paper [30] which fully reflects the commonly accepted status of this field.

To conclude, I will be able to recommend publication of this manuscript after the authors implement all necessary modifications suggested under (1), (2) and (3) and take into account my general remarks under (4).

Reviewer #3 (Remarks to the Author):

The authors have already addressed all my questions and comments. This work is novel and interesting, therefore, I recommend it to be published in Nature Communications.

Reply to the second reviewer

I have carefully read through both the revised version of the manuscript and the authors' rebuttals to my previous remarks. At a number of points the authors' responses are satisfactory. At those instances they improved both the presentation and the bibliography in a proper manner. Still, there remain a few issues where their text revisions and comments come incorrect and/or misleading.

We thank the Referee for his detailed and critical reading of our manuscript. We find that most of his remarks are well taken, while some apparent disagreement is only due to a misunderstanding between us and the Referee, that we clear up below.

Before giving detailed answers to the referee's comments, we would like to emphasize an important point: the three devices presented in our study show a Josephson phase localized in one well due to the Schmid transition, since the effective environmental impedance (as seen by the Josephson junction) is close to $4 \text{ k}\Omega$, which is smaller than the resistance quantum.

1. In response to my critique in the Introduction section the authors have added the sentence: «Provided that is sufficiently smaller than π^2 , $\propto Z_J/R_Q$. ». While the comment on the validity range of the above estimate for is indeed appropriate, the indicated condition $\ll \pi^2$ is not quite correct. The estimate $\propto Z_J/R_Q$ actually holds only provided remains (much) smaller than unity, whereas the number $\pi^2=10$ is totally irrelevant in this context. The authors should keep the above sentence in place but correct it accordingly, e.g., simply replacing « π^2 » by the word «unity».

We have reconsidered the offending statement. A condition on $\langle \phi^2 \rangle$ may be obtained, by demanding that the overlap of two Gaussian wave functions separated by 2π be small. The overlap between these wave functions is given by $\exp[-\pi^2/2 \langle \phi^2 \rangle]$, hence the criterion $\langle \phi^2 \rangle \ll \pi^2/2$, although this may be a matter of taste. We are happy to change this sentence in the introduction as the referee requests to $\langle \phi^2 \rangle \ll \text{unity}$.

2. Below Eq. (2) the authors have added the sentence: «This is valid, provided is sufficiently less than π^2 . » Furthermore, in the section «Quantum versus thermal fluctuations» below Eq. (6) they have added the paragraph: « We also note that at low temperature the largest phase fluctuations are approximately ~ 1.3 , which is significantly less than π^2 , which justifies the use of the SCHA. A likely source of the discrepancy between theory and experiment at high temperatures, is the breakdown of the SCHA due to temperature enhanced phase fluctuations... ». These text additions demonstrate that the authors do not fully understand the method of SCHA in general and its applicability range in particular. SCHA is a variational technique replacing an anharmonic potential by an effective harmonic one with some effective parameters to be determined self-consistently. Hence, within the framework of SCHA in Eq. (2) can in principle take ANY value ranging from zero to infinity. If, for instance, the latter limit is realized, i.e. tends to infinity, it would simply imply that (within SCHA!) the effective Josephson coupling energy E^*J gets renormalized down to zero by interactions. Hence, no ad hoc restrictions on exist within SCHA and the sentence right after Eq. (2) as well as the cited paragraph below Eq. (6) should be deleted as misleading. Perhaps I can also add that even following the authors' logics the discrepancies observed in Fig. 3b at higher temperatures could hardly be explained since for all three samples remains much smaller than 10 up to

We agree with the referee that the SCHA can yield as a solution a Gaussian with arbitrarily large width, or equivalently a zero effective Josephson coupling E^*J . However, whether this result is to be trusted depends on the question being investigated. We elaborate on when the SCHA gives an accurate estimate for the renormalized weak link resonance frequency below, in our response to the fourth point that the referee raises in her/his second report. There, we also show that reasonable results can still be obtained for $\langle\phi^2\rangle$ around 1. Regarding larger temperatures, it is indeed puzzling at first sight that the agreement between theory and experiment deteriorates so significantly when the extracted $\langle\phi^2\rangle$ only changes from 1.3 to 2.25. The likely resolution is a systematic underestimation $\langle\phi^2\rangle$ in the extraction process, when $\langle\phi^2\rangle$ is relatively large. This is because at large $\langle\phi^2\rangle$, the reduction of ω_J^* that we measure with increasing T corresponds to a bigger increase in $\langle\phi^2\rangle$ than our formula predicts. We explain this more explicitly in the new manuscript.

3. Towards the end of the Introduction section the authors added the sentence «In DC measurements, such effects result in the celebrated Schmidt-Bulgadaev transition predicted more than thirty years ago [27, 28], a localization phenomenon whose relevance for microwave AC measurements is presently debated [29]». The first part of this sentence is OK (apart from misspelling the name Schmid), while the second one (quoting a highly controversial preprint [29]) is problematic, since BY DEFINITION any quantum phase transition (QFT) including, of course, the localization QFT of the Schmid-Bulgadaev type, deals with the limit of both zero temperature and zero frequency. Hence, there is nothing to debate. I would strongly recommend the authors to leave out the second part of the above sentence in order to avoid any mentioning of microwave AC measurements in the context of the Schmid-Bulgadaev QFT.

We agree with the Referee that the interpretation of the experiment of Reference [29] is controversial, and that a proper low frequency analysis was not performed by these authors. We have thus corrected our sentence.

4. The authors' reply to the comment #5 from my previous report is by no means satisfactory. In addition to an improper discussion of the SCHA validity range (already mentioned above) this reply contains a number of incorrect and misleading statements which I cannot avoid commenting on. For instance, the authors claim that «...at equilibrium the superconducting phase remains compact...». Or, they write that «The estimate $=\sqrt{2 E_c/E_{\{J,\{\rm bare}\}}}$ is well-known in the transmon qubit literature... and holds quite well down to $E_J/E_C=1$, as can be verified from the exact solution for this problem». Unfortunately, these and some other related statements are in error. To begin with, the issue of being in or out of equilibrium has nothing to do with compactness or non-compactness of the superconducting phase. The latter is compact as long as the Josephson junction (JJ) charge remains discrete (e.g., quantized in units of $2e$) and gets decompactified if the charge of JJ becomes effectively continuous as a result of interactions with an external circuit. Furthermore, even if the phase is compact (i.e. defined on a ring), it does not imply that it is necessarily confined to the interval between zero and 2π , contrary to the authors' belief. The point is that the phase can take as many winds (rotations) around the ring -- both clockwise and counterclockwise -- as possible. As a result, e.g., in the true ground state of an isolated JJ and for ANY relation between EJ and EC the charge is always localized (thus being a c-number variable), whereas the phase suffers strong quantum fluctuations making diverge. It is just THIS behavior which follows from the exact solution of the problem no matter what «is well-known in the transmon qubit

literature». In other words, an isolated JJ in or close to its ground state always demonstrates Coulomb blockade of Cooper pairs, thus being an insulator rather than a superconductor. In fact, this simple physics was well established and understood already three decades ago. For more details I can refer the authors, e.g., to the review paper [30] which fully reflects the commonly accepted status of this field.

The fourth point the referee raises is that our response to her/his point 5 in the previous report is not satisfactory. Before responding, let us collect the relevant comments from both reports.

The following comes from the referee's first report. (We use our own numbering)

1. *In the introductory part of the manuscript the authors give a simple estimate for the averaged square of the phase fluctuations as being proportional to the ratio ZJ/RQ . Note that this estimate may only be appropriate under two conditions: (*) provided the Josephson phase is localized to one well, e.g., due to the so-called Schmid dissipative phase transition and, on top of that (*) provided the strong inequality $ZJ \ll RQ$ is satisfied.*
2. *The condition (*) is not discussed by the authors at all.*
3. *Moreover, in the text between Eqs. (14) and (15) the authors argue that their estimate for the averaged square of the phase fluctuations holds for an isolated JJ. This statement is fundamentally flawed since in equilibrium such JJ behaves as an insulator due to Coulomb blockade of Cooper pairs. Hence, the Josephson phase fluctuates strongly and the authors' estimate for phase fluctuations does not apply even qualitatively. Such fluctuations can strongly be reduced only in the presence of a low-impedance environment which crucial role appears to remain unclear to the authors.*

This should be read in conjunction with the following remarks from his second report. (Our numbering again.)

4. *The authors do not fully understand the method of SCHA in general and its applicability range in particular.*
5. *As a result, e.g., in the true ground state of an isolated JJ and for ANY relation between EJ and EC the charge is always localized (thus being a c-number variable), whereas the phase suffers strong quantum fluctuations making $\langle \phi^2 \rangle$ diverge. In other words, an isolated JJ in or close to its ground state always demonstrates Coulomb blockade of Cooper pairs, thus being an insulator rather than a superconductor.*

In points 1, 2 and 3 above, the referee says that unless you have a low impedance environment shunting the junction, that brings about a Schmid transition that localizes the phase, the phase has diverging fluctuations (and charge uncertainty is small, which is synonymous with insulating behavior). These remarks likely refer to the following section of the Schoen Zaikin review:

At $T=0$ in junctions with purely Ohmic dissipation the result is summarized in the phase diagram fig. 17. For $\alpha_s < 1$ the phase φ fluctuates strongly and the average $\langle \varphi^2 \rangle$ diverges, while for $\alpha_s > 1$ the phase φ is localized ($\langle \varphi^2 \rangle$ is finite and does not increase in time) and we can regard it as a classical variable. Thus the junction shows superconducting properties provided Ohmic dissipation is strong, $\alpha_s > 1$. This answer may appear paradoxical in two respects. First, we see that there is no superconductivity in Josephson junctions without dissipation, or, in other words, we need dissipative currents to have superconductivity. However, this result is consistent with the general observation that dissipation suppresses quantum fluctuations. Second, the phase boundary between the superconducting ($\alpha_s > 1$) and the normal ($\alpha_s < 1$) parts of the phase diagram of fig. 17 does not depend on the ratio E_J/E_C .

The absence of superconductivity for weak dissipation can be easily understood in the limit $E_J \ll E_C$. We can roughly estimate the uncertainty in φ from the relation (see ref. [1]) $\delta Q \delta \varphi \approx e$. At low temperatures $k_B T \ll E_C$ the junction charge is fixed ($\delta Q \rightarrow 0$) and thus quantum fluctuations of φ are strong. The opposite case $E_J \gg E_C$ is less trivial. At first sight in this limit the phase should be localized near the bottom of the potential well (say, near $\varphi = 0$) and quantum fluctuations of φ should be small. Consequently, one can expect the existence of classical Josephson effects for $E_J \gg E_C$ even for small α_s . Moreover, a huge number of experiments where Josephson junctions with large values of $E_J \gg E_C$ and $\alpha_s < 1$ show classical behavior appear to contradict the claimed absence of superconductivity in this parameter range. However, closer inspection shows that there is no contradiction between all these facts. We have argued (see section 3.3) that in many circumstances φ should be regarded as an extended variable $-\infty < \varphi < \infty$, that is it plays the role of the coordinate of the quantum particle in a periodic potential $E_J \cos \varphi$. Even if φ is initially localized in one well it will spread due to quantum tunneling to the vicinity of the points $\varphi = 0, \pm 2\pi, \dots$ and thus for long times $\langle \varphi^2 \rangle \rightarrow \infty$. The large uncertainty in φ allows sharp values of Q . But if the junction charge is fixed the whole current (for small I_x) will flow through the shunt (Coulomb blockade) and the system has no superconducting response.

In the above paragraph, the discussion of the $E_J \gg E_C$ regime is particularly important to understand what the referee is saying. Basically, the argument is this. When $E_J \gg E_C$, one could be forgiven for thinking that $\langle \varphi^2 \rangle$ is small, because it is localized near the minimum of the cosine potential. However, $\langle \varphi^2 \rangle$ actually diverges because φ is equally likely to be in any unit cell of the cosine potential, even though within a given unit cell, it is concentrated close to the minimum of the cosine potential.

Our response to the referee on this is the following:

Firstly, we apologize for our lack of clarity and for using sentences such as “at equilibrium the superconducting phase remains compact”. We were trying to make a point about the absence of terms in the Hamiltonian that explicitly break 2π translation symmetry, but in hindsight appreciate that it is not very relevant, and easily misconstrued. We don’t rely on the argument we were trying to make previously any more. Secondly, as mentioned earlier in this reply, our three devices are operated in the “localized phase” part of the Schmid-Bulgadaev diagram ($2Z_{\text{env}} < R_Q$). We apologize for the oversight on our part for not making this very clear in our previous response. It is common cause between us and the referee that this validates the use of the SCHA.

The above point makes the rest of the discussion below somewhat moot, in the sense that it does not matter whether the SCHA can be applied in the case of an isolated junction. However, to assure the editor and the referee that we have taken the same care to respond to the referee’s concerns, as the referee has taken in evaluating our work, let us explicitly address the interesting point regarding isolated junctions that has been raised.

As mentioned in the part of the Schoen Zaikin review that we quote above, the wave function of the isolated junction is 2π periodic in φ (if Q_x is zero). In this case the SCHA gives reasonable results when the wave function consists of identical narrow peaks a distance 2π apart as φ is varied. In this case the SCHA consists of finding a reasonable Gaussian line-shape for an individual peak. A necessary condition, is that the Gaussian we find must have already decayed to a small value in the vicinity of the next peak. This is

because we approximate the ϕ integral of the true energy density over a 2π interval centered around one peak, by the ϕ integral from minus infinity to infinity of the energy density of the single Gaussian peak.

When we talk about fluctuations of ϕ , we refer to the uncertainty in ϕ within one 2π period, centered around a peak in the wave function. This uncertainty is finite and tells us how broad one peak of the wave function is. It contains information on the weak link's resonant frequency, because it tells us about sloshing motion about a single minimum of the potential. This is to be contrasted with the fluctuations associated with phase tunnelling events between different unit cells of the cosine potential. The latter diverges and is the appropriate quantity to diagnose charge localization (i.e. insulating behavior). When the SCHA is accurate, the uncertainty of ϕ in the unit cell is approximately equal to the width of the Gaussian that approximates an individual peak. Let us discuss these issues further below, by studying an explicit, although simplified, example.

The isolated Josephson junction system that we have in mind, corresponds to Eq. 2.19 in the review paper, with one change of convention, namely E_C in our manuscript = 4 times E_C in the review. We will use our definition of E_C in the discussion, and consider the point $E_{JQ}/E_C = 1$. The left panel in the figure above shows the optimal Gaussian approximation for a single well as well as the exact wave function for offset charge $Q_x=0$. Also shown on the right panel is the exact energy density (the ground state energy is the area under the yellow curve from $-\pi$ to π in the right panel), versus the variational estimate (the variational energy is the area under the blue curve, from $-\infty$ to ∞). We see quite good agreement. Indeed, as the offset charge Q_x sweeps over one Cooper pair, the exact ground state energy sweeps over the interval $0.88 E_0 - 0.92 E_0$, while SCHA as we implement it, gives $0.91 E_0$. Here $E_0 = (E_C E_{JQ}/2)^{1/2} = E_{JQ}/\sqrt{2}$. Note that the constant wave function, i.e. the trial state for $E_{J^*}=0$, gives a variational energy E_{JQ} , that is higher than the non-trivial solution.

Let us now see what happens when we decrease E_{JQ} further. For $E_{JQ} < 0.34$ the constant trial state has a lower energy than that of the non-trivial trial state with nonzero E_{J^*} . However, this does not signal any kind of real phase transition, but rather the complete breakdown of the SCHA method, due to spurious energy contributions from the ϕ intervals $(-\infty, -\pi] \cup [\pi, \infty)$.

We note that for the isolated junction studied above, $E_{JQ}=E_C$ is close to the lowest value of E_{JQ} for which we are prepared to trust the SCHA. For smaller values of E_{JQ} , a single peak of the true wave function soon becomes too fat to be well-approximated by a Gaussian than neatly fits into a single 2π unit cell, and is small outside it. However, when

an environment is present, its damping effect leads to narrower individual peaks in the wave function for a given value of E_{JQ}/E_C , than in the isolated system. To see if the optimal trial states that we obtain for the full weak link + environment system meets the requirement of neatly fitting into a single ϕ unit cell, we can compute the probability to find ϕ in $[-\pi, \pi]$ for a given optimal Gaussian trial. A necessary condition for the applicability of our approximation scheme is that this probability must be close to unity. For the above example of an isolated junction with $E_{JQ}=E_C$, one finds a probability of 99.9%. For our sample A at zero temperature and $E_{JQ}=0.25 E_C$, the probability is 99.4%. In other words, due to environmental damping, the SCHA is only a slightly worse approximation for sample A, despite its low E_{JQ}/E_C ratio, than it is for the isolated junction we discussed above, for which it does a very reasonable job.

To conclude, I will be able to recommend publication of this manuscript after the authors implement all necessary modifications suggested under (1), (2) and (3) and take into account my general remarks under (4).

We thank again the referee for raising these important questions. We hope we've addressed her/his concerns appropriately.

We have modified our manuscript according to referee's suggestions:

- In response to point one of the referee's second report, we have changed the sentence in the introduction (second column, page 1) from "Provided that $\langle \phi^2 \rangle$ is sufficiently smaller than π^2 ..." to "When $\langle \phi^2 \rangle$ is sufficiently less than unity ..."

-In response to point 2 of the referee's second report, we have firstly removed the sentence "We also note that at low temperature the largest phase fluctuations are approximately $\langle \phi^2 \rangle \sim 1.3$, which is significantly less than π^2 , and justifies the use of the SCHA." below Equation 6. Secondly, Further down the same paragraph, we have replaced the sentence "However, the true fluctuations of the phase in the circuit rapidly become larger than the extracted values (possibly higher than the maximal value of 2.25 that was estimated for $\langle \phi^2 \rangle$), leading for sample A to some mismatch with the theory at the highest temperatures." with "It is likely that the extracted $\langle \phi^2 \rangle$ for sample A is systematically underestimated due to sizable errors in the SCHA that rapidly set in after $\langle \phi^2 \rangle > \sim 1$, leading to a mismatch with the theory at high temperatures."

-In response to point 3 of the referee's second report, we have made the following change: When mentioning the Schmidt-Bulgadaev transition and the reference [29], we now write "a localization phenomenon whose relevance for microwave AC measurements requires further experimental and theoretical investigations".

- In response to point four of the referee's second report, we have added the following text in the paragraph above equation 2: "This is valid, provided the phase ϕ , though strongly fluctuating, is still sufficiently localized. In this regard we note the following. Though large, the effective environmental impedance $2 Z_{\text{chain}} \approx 3.8 \text{ k}\Omega$ seen by the weak link, is still less than R_Q . Under this condition, the environment is known to produce spontaneous symmetry breaking of the 2π periodicity in the phase difference ϕ across the weak link, [19, 20, 22]. It is therefore reasonable to approximate the system's full wave function with a Gaussian that is fairly well localized in the ϕ direction, which is the essence of the SCHA." We have also removed the sentence "This is valid, provided $\langle \phi^2 \rangle$ is sufficiently less than π^2 ." below Eq. 2.